# Visualizing acyl carrier protein interactions within a crosslinked type I polyketide synthase

Ziran Jiang[1,2], Graham W. Heberlig [1,2], Jeffrey A. Chen [1], Jennifer Huynh [1], James J. La Clair [1] & Michael D. Burkart [1] ✉

Using a combination of dual covalent crosslinking and cryo-EM analyses, we elucidate the structure of mycocerosic acid synthase from *Mycobacterium tuberculosis* trapped in two distinct catalytic states during its iterative cycle. These structures reveal domain architecture of the acyl carrier protein mediating condensation and dehydration through dual site-selective crosslinking of the acyl carrier protein with the ketosynthase and dehydratase domains. Map density was sufficient to visualize full domain architecture with active site-bound probes and elucidate key interactions of four distinct crosslinked species. Here, iterative vectorial polyketide biosynthesis arises through an overall twisting and tilting architecture, enabling positioning and entry of the cognate substrate at each enzymatic domain. These structures present valuable details for future therapeutic design against mycocerosic acid biosynthesis in *M. tuberculosis*.

Approximately 25% of the global population is infected by *Mycobacterium tuberculosis* (Mtb), responsible for 1.5 million annual deaths worldwide[1] and a leading cause of child mortality[2]. This has been aggravated by multidrug-resistant strains, which now infect over half a million patients, and new therapies will be crucial for future control[3,4]. As exemplified by isoniazid, which targets the enoylreductase in mycolic acid biosynthesis, mycocerosic acid synthase (MAS) offers a potential drug target (Figs. 1a and 2b) by producing part of phthiocerol dimycocerosate (PDIM) (Supplementary Fig. 1), a key component of the cell envelope[5]. While PDIM makes up 46% of extractable Mtb lipids, it plays a major role in Mtb virulence, immunosuppression, and evasion[6–8]. MAS, a fully reducing iterative type I polyketide synthase (PKS), is composed of six domains (Fig. 2a, b) that together catalyze the elongation of *n*-C6-C20 fatty acids to prepare mycocerosic acid[9,10]. Within this 448 kDa obligate homodimer, five monofunctional enzymatic domains sequentially catalyze discrete chemical steps upon a growing metabolite that is covalently tethered to the acyl carrier protein (ACP)[5] (Fig. 1a and Supplementary Fig. 1).

Type I PKSs, including MAS, have garnered significant attention as potential therapeutic targets and for the rational design of metabolites[11].

However, progress toward their understanding has been hindered by the lack of fundamental structural knowledge attributed to their inherently large size and complex structural dynamics. Early type I PKS structural studies focused on excised catalytic domains, didomains, or pruned modules[5,12], where the ACP is either missing or poorly resolved. As single particle cryo-EM has advanced, modular PKSs have been captured utilizing antigen-binding fragments for structural stabilization, enabling some ACP visualization in complex with select domains, but the map density of substrates and ACP structure has remained missing or fragmentary[13–17].

To address these challenges, we developed dual site-selective crosslinking probes as tools to conformationally constrain PKSs in their functional states for structural analysis[18]. Tethering a covalent crosslinker onto the ACP can trap the interactions mediating catalysis when crosslinked with catalytic domains, offering unambiguous structural information of the protein-protein interactions (PPIs) and protein-substrate interactions[19]. We aim to visualize catalytic domains in complex with ACP and chose MAS as a type I PKS model by which to develop and demonstrate these tools (Fig. 1b–d). Here, we elucidate the complete structure of MAS with well-defined ACP interactions

[1]Department of Chemistry and Biochemistry, University of California, San Diego, La Jolla, CA, USA. [2]These authors contributed equally: Ziran Jiang, Graham W. Heberlig. ✉e-mail: mburkart@ucsd.edu

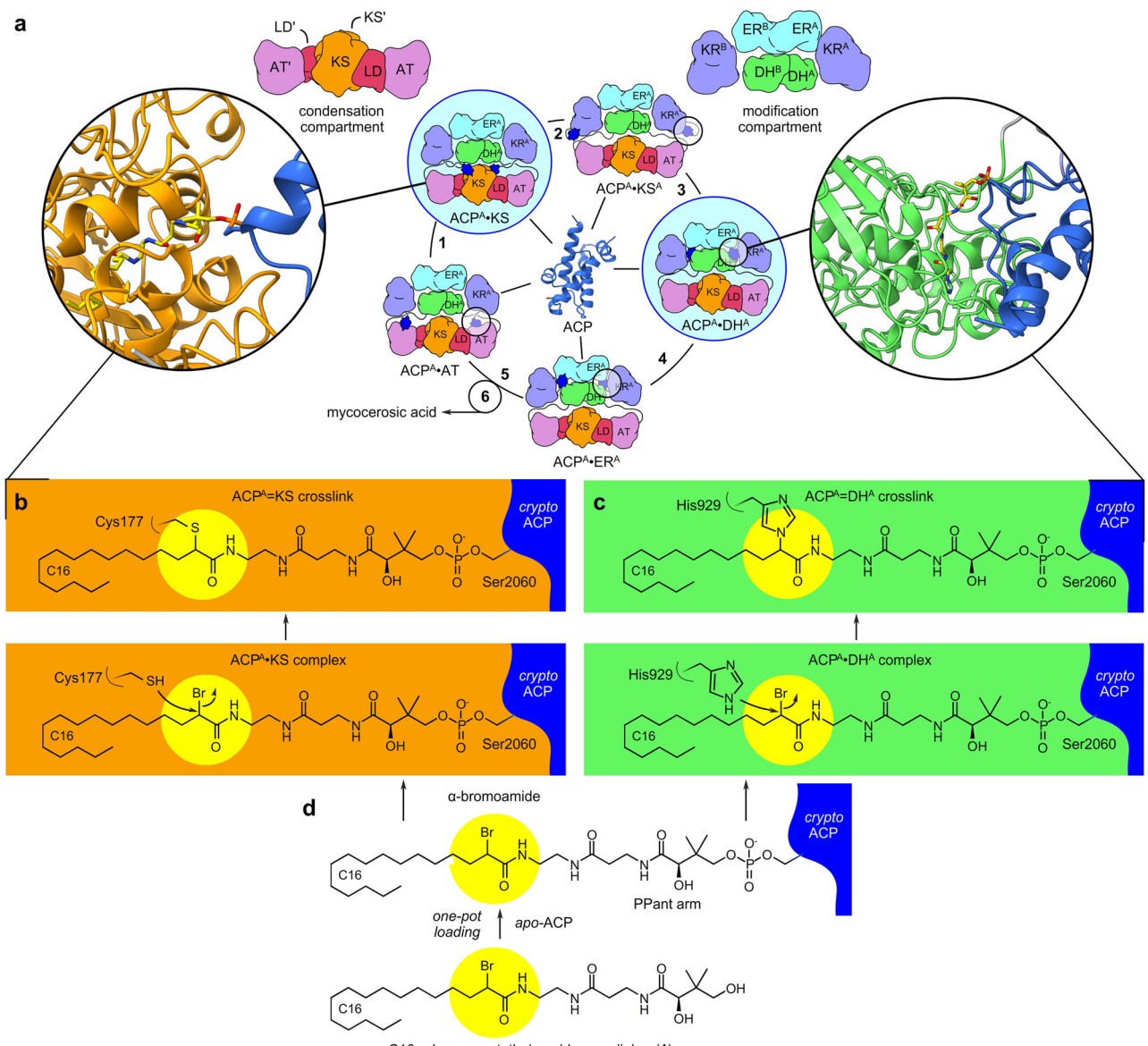

**Fig. 1 | Dual site-selective crosslinking of the mycocerosic acid synthase (MAS).**
**a** Depiction of ACP processivity in MAS biosynthesis. ACP must interact with each catalytic domain, acyltransferase (AT, step 5 extension unit loading), ketosynthase (KS, step 1 condensation. Substrate loading or reloading step is not shown), ketoreductase (KR, step 2 β-ketoreduction), dehydratase (DH, step 3 dehydration), and enoylreductase (ER, step 4 α,β-enoylreduction), through discrete protein-protein and protein-substrate interactions during each catalytic cycle. MAS can be divided into two compartments, one that serves to load and elongate acyl chains (condensation compartment, steps 1 and 5); and a second that serves to modify and fully reduce the condensation product (modification compartment, steps 2–4). MAS exists as a homodimer, where protomer 1 (indicated by [A] after each abbreviation) is depicted with domain abbreviations, and protomer 2 (indicated by [B] or by prime' after each abbreviation). **b** Mechanism of ACP = KS (=indicates crosslink) crosslinking reaction. **c** Mechanism of ACP = DH crosslinking reaction. **d** This process is achieved through chemoenzymatic loading of **1** onto the *apo*-ACP to deliver *crypto*-ACP. Once loaded, the active site Cys of KS or His of DH react with the α-Br-pantetheinamide warhead, yielding a crosslinked complex.

through the application of dual site-selective crosslinking (Fig. 2a and Supplementary Fig. 2) and single particle cryo-EM analysis, resolving key protein-protein interactions between the ACP with enzymatic domains and linker regions that to date have remained unresolved. We find that dual covalent crosslinkers combined with cryo-EM single particle analysis can offer access to multiple ACP bound states and provide a dynamic framework from which to visualize complex ACP interactions in multidomain synthases.

## Results
### Domain-selective crosslinking of MAS
We have previously demonstrated that dual covalent crosslinkers (Supplementary Fig. 3) efficiently trap the ACP with a variety of partner

domains, including ketosynthases (KS) (Fig. 1b)[19,20], dehydratases (DH) (Fig. 1c)[21,22], and other partner proteins[23,24] in type II fatty acid synthase (FAS) and PKS systems. Here, a panel of crosslinkers (Supplementary Fig. 3) can be loaded onto an ACP through a chemoenzymatic one-pot reaction using four enzymes (CoaA, CoaD, CoaE, and Sfp) to access *crypto*-ACP (Fig. 1d)[25,26]. In type II systems the *crypto*-ACP can be generated separately, purified, and incubated in any desired molar ratio with the target partner protein in trans (intermolecularly between two proteins). However, when applying this technique to type I PKSs, where multiple domains are found within a single megasynthase protein, the free reactive crosslinker has the potential to react with an enzymatic domain, thereby inactivating it and creating shunt domain products (Supplementary Fig. 4a).

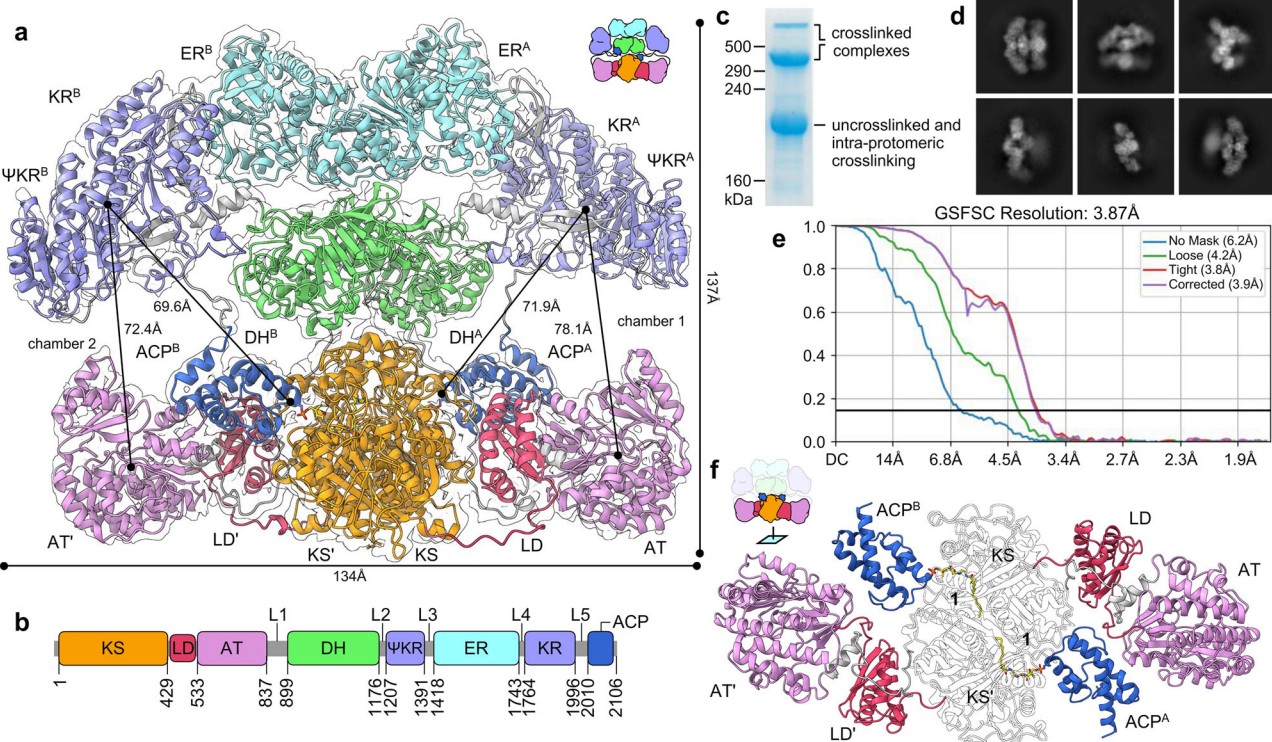

**Fig. 2 | Architecture of complex A displaying crosslinking of ACP = KS domains.**
**a** Structure of complex A at 3.9 Å with a contour level of 0.088. Distances of key residues in select domains are shown to compare the conformation of each protomer. The homodimeric architecture provides two catalytic chambers housing the catalytic process. ACPs engage their catalytic domains in chambers 1 and 2. **b** Linear organization of each domain in respective MAS protomers, where L1-L5 are linkers 1 to 5. Two protomers comprise the structure in (**a**). **c** Coomassie brilliant blue stained 6% SDS-PAGE gel (Supplementary Fig. 45) depicting the outcome of crosslinking with **1**. **d** Selected particle stacks from 2D classification. **e** FSC curve for complex A density map (Supplementary Figs. S8–S9). **f** Demonstration of ACPs crosslinked at KSs, revealing dual site-selective crosslinking with **1** (Fig. 1b).

To avoid such shunt products, we first attempted to crosslink (Supplementary Fig. 5) the excised *crypto*-ACP domain with excised KS, excised KS-LD-AT (including linker domain LD) of MAS, or *E. coli* FabB (type II FAS KS) in trans with a panel of crosslinkers (Supplementary Fig. 3). This experiment was designed to also investigate the PPIs between excised MAS ACP and excised MAS KS through structural analysis. However, modest or no crosslinking yield was observed between the *crypto*-ACP and KS or KS-LD-AT (Supplementary Figs. 5a, b). This is likely due to inactivation of the KS, as it has been reported that the excised MAS-like PKS KS-LD-AT tridomain construct exists as a monomer-dimer equilibrium in solution[5]. This interpretation is supported by published observation of KSs in both type I and type II systems, confirming that KS domains function as obligate homodimers. Surprisingly, slight crosslinking between FabB and excised MAS-ACP was observed, indicating that MAS-ACP is functional even though excised. The low crosslinking yield was most likely due to the difference in ACP-KS PPIs between *E. coli* type II FAS and *M. tuberculosis* type I PKS systems. We examined crosslinking between the excised MAS ACP and intact full-length MAS. With the intact MAS, *crypto*-ACPs loaded with a panel of probes crosslinked efficiently (left, Supplementary Fig. 5c), confirming that the KS activity requires a stable dimerization interface provided by other domains such as DH and ER. In comparison, crosslinking between *E. coli* AcpP (right, Supplementary Fig. 5c) and intact MAS showed only very limited crosslinking with the same panel, demonstrating that PPIs between ACP and KS domains are distinct in type I and type II systems. These studies of crosslinking in trans demonstrate the importance of strict PPIs and domain boundaries as exemplified in previous PKS engineering works[27,28].

## Crosslinking enables cryo-EM structure elucidation of multiple complexes

Unproductive crosslinking studies with excised MAS domains in trans led us to explore the reaction *in cis* (intramolecularly within the same protein) under the hypothesis that crosslinking between the ACP and KS could enhance resolution in cryo-EM by rigidifying these dynamic multidomain proteins (Fig. 1 and Supplementary Fig. 2). To evaluate MAS with the ACP crosslinked to KS (Fig. 1b), we revisited our methodology to improve crosslinking probe loading efficiency. We first converted the probes into the primed CoA analogs, followed by addition of *apo*-MAS to maximize the yield of crosslinked MAS (Supplementary Fig. 4b). In this way, this optimized reaction sequence minimized interaction time between the free probe and the *apo*-MAS, limiting shunt product formation.

We then turned our attention to selecting an optimal probe for crosslinking MAS *in cis* (Supplementary Fig. 3). With in trans crosslinking in *E. coli* ACP as a model (Supplementary Fig. 6), we learned that α-bromoamide probes can target both KS and DH enzymes. Here, the use of a probe that targets more than one domain could provide a useful tool to trap the dynamic multistate movement in MAS (Fig. 1a). We selected C16-α-bromoamide **1** (Figs. 1b–d and Supplementary Fig. 3), where a long lipid tail mimicked the growing mycocerosate chain, and the α-bromoamide moiety offered excellent crosslinking efficiency (Fig. 2c). Size exclusion chromatography afforded crosslinked MAS products at >45% crosslinking yield by SDS-PAGE (right, Supplementary Fig. 7). Aliquots of a 1 mg/mL sample were applied to holey carbon grids, blotted, and plunge-frozen in liquid ethane on a Vitrobot Mark IV. Micrographs were collected at 130,000× nominal magnification with a pixel size of 0.889 Å and

exposure dose of 55 e⁻/Å² on a Titan Krios G4 electron microscope. The resulting movies were processed within Cryo-SPARC and curated to remove micrographs with poor CTF fit, large in-frame motion, and images with overly thick ice. After unsupervised particle picking and filtering (Supplementary Figs. 8–9), an initial stack of ~1,050,000 particles was obtained and further classified, filtered, and refined to generate four reconstructed maps for complexes A-D from a single data set. In complex A (Supplementary Fig. 2), all domains KS, LD, AT (acyl transferase), DH, ΨKR (nonfunctional ketoreductase as structural support), ER (enoylreductase), KR (ketoreductase), and ACP are visualized. Both ACPs crosslinked at KSs with probe **1**. Based on the SDS-PAGE results of crosslinking, cross-protomer crosslinking was observed, but the data does not exclude crosslinking within the same protomer (Supplementary Figs. 7 and 45). Complex B (Supplementary Fig. 2) revealed the homodimeric condensation compartment (KS-LD-AT) with one ACP crosslinked at KS via **1**, while the other KS was inhibited by **1** in small molecule form (without ACP). Complex C (Supplementary Fig. 2) demonstrated the modifying compartment (DH-ΨKR-ER-KR), with ACP crosslinked with DH through **1**. Complex D (Supplementary Fig. 2) revealed all domains, with both ACPs crosslinked at DH and KS, respectively.

Evaluating these four crosslinked complexes offered several important structural insights. First, dual site-selective crosslinking enabled high-resolution visualization of multidomain synthases detailing MAS ACP interactions with crosslinked domains, adjacent domains, and linker peptides. Second, ACP crosslinking can rigidify MAS, providing higher local resolution at the crosslinked sites. Finally, dual site-selective crosslinking greatly enhanced the homogeneity of 4 trapped states within 6 catalytic states in each catalytic chamber.

### Architecture of the ACPs = KSs dual crosslinked complex A

We obtained a density map with 3.87 Å resolution (Fig. 2) from samples of the ACPs = KSs ( = indicates crosslink) dual crosslinked protein, complex A, including all domains KS, AT, DH, ΨKR, ER, KR, and ACP (Fig. 2a and Supplementary Figs. 10–12). Both ACPs are crosslinked at the homodimeric KS domains, linking Ser2060 in ACP with Cys177 in KS with the C16 acyl tail bound within the canonical KS substrate pocket (Fig. 2f). Our SDS-PAGE results (Supplementary Figs. 7 and 45) demonstrate cross-protomer crosslinking, but they do not exclude crosslinking within the same protomer. The full MAS structure adopts a homodimeric architecture (Fig. 2a), comprising two catalytic chambers (chambers 1 and 2, Fig. 2a). The dimerization interface is comprised of the KS • KS' (2566 Å²), DHᴬ • DHᴮ (429 Å²), and ERᴬ • ERᴮ (1165 Å²) domains. The condensation compartment consists of a KS • KS' homodimer core followed by two AT domains on opposite sides spaced out by LD domains (Supplementary Fig. 13a). While the LD domain acts as a spacer between the KS and AT domains, it has limited contact with the KS. Intriguingly, the 62 residue post-AT linker (L1) brings the KS and AT into proximity via a polar interface (Supplementary Fig. 13a). L1 wraps around the LD, forming an LD/L1 interface (1627 Å²), which offers extensive L1/KS (970 Å²) and L1/AT (320 Å²) interactions. The LD has been previously reported to be involved in a turnstile mechanism, where in a closed conformation it forms extensive interactions with the AT to block access to the KS[13]. While such a turnstile-closed state ensures vectorial biosynthesis in the DEBS type I modular PKS system[13,29], it is not observed in complex A, which is an iterative type I PKS (Supplementary Fig. 13).

From the AT, L1 leads to the DHᴬ • DHᴮ of the modifying compartment which resides just above the KS • KS' core (Supplementary Figs. 13a, 52a, and 52b). Residues 882-893 residing in C-terminal of linker 1 were not modeled in due to weak density. Due to this, the connection between the condensation and modifying compartments cannot be defined within the same protomer. As a result, two separate labeling systems of KS-LD-AT/KS'-LD'-AT' for the condensation compartment and DHᴬ-ΨKRᴬ-ERᴬ-KRᴬ-ACPᴬ/DHᴮ-ΨKRᴮ-ERᴮ-KRᴮ-ACPᴮ for

the modifying compartment are used. DHᴬ • DHᴮ is a dimer of two double-hotdog folds, forming a "handshake fold" and creating a 429 Å² interface[5]. The catalytically inactive pseudo-KR domain (ΨKR) is linked to the DH through a 31 residue DH-ΨKR linker (L2) (Supplementary Figs. 13b and 52a, b). Next, a 27 residue linker 3 (L3) connects the ΨKR to the ER, which is a homodimer that forms a 1165 Å² interface, followed by a 21 residue linker 4 (L4) connecting ER to KR. Altogether, L2, L3, and L4 form two sets of antiparallel β-strands, providing extensive interfaces with the KR (2365 Å²) and ER (686 Å²) domains. Supported by partial density, a 14 residue flexible KR-ACP linker (L5) allows the ACP to travel between the domains of each of the catalytic chambers of MAS (Supplementary Figs. 13b and 52a, b). The linker-based network creates two functional catalytic chambers and ensures flexibility for the ACP to reach all domains akin to porcine FAS (Supplementary Figs. 48 and 49). Due to the flexibility offered by the linker-based architecture, complex A is not perfectly symmetrical. Distances between the active site residues Tyr1912 (KR) and Ser623 (AT), as well as Tyr1912 (KR) and Ser2060 (ACP), are 78.1 Å and 71.9 Å, respectively, in protomer 1, compared with 72.4 Å and 69.6 Å inter-residue distances in protomer 2, revealing a significant in-plane tilting movement of the modifying compartment that pivots at the KS • KS' core (Fig. 2a). The double crosslinking between ACPs and KSs was supported by unambiguous map density. The crosslinkers tethered on the ACPᴬ and ACPᴮ through a 4'-phosphopantetheine (PPant) arm are well resolved within the KS • KS' binding tunnels, each making a covalent linkage at the active site Cys residues (Cys177 in KS and KS') (Fig. 2f and Supplementary Fig. 14).

### Intramolecular Crosslinking Between ACP and KS in Complex B

We also obtained a density map for complex B (Fig. 3 and Supplementary Fig. 15) from the same dataset with a higher overall resolution (3.22 Å) that further elucidates details of the condensation compartment (KS-LD-AT), with only one ACP crosslinked at KS (Fig. 3a and Supplementary Figs. 15–17). Due to the structural rigidification of covalent ACP crosslinking, the local resolution around the ACP crosslinked region is higher than the non-crosslinked region (Supplementary Fig. 15). This effect is particularly pronounced in AT' in complex B. Such a rigidification effect through ACP crosslinking binding was also observed in the recent DEBS structures[30]. Along with crosslinker 1, ACP was unambiguously resolved. Covalent bond S-C between the KS active site Cys177 and the reactive α-position of the crosslinker (Fig. 3e and Supplementary Fig. 18), as well as the O-P covalent bond between the ACP active residue Ser2060 and the PPant arm (Fig. 3e and Supplementary Fig. 18), were determined by continuous map density. In the other protomer, continuous density was observed only for crosslinker **1** reacted with Cys177 of KS but not ACP due to the shunt reaction of **1** (Supplementary Fig. 19).

In MAS, the ACP approaches the KS domain with a head-on orientation (Fig. 3a and Supplementary Fig. 20A). This contrasts with type II KS-ACP interactions, typified by AcpP = FabB (PDB ID: 7L4L), where the AcpP α-helical bundle is tilted to allow the KS to make more contact with helices 1, 2, and 3 (Supplementary Fig. 20d) and loop 1 of ACP. Type II ACPs are also typically negatively charged and utilize extensive electrostatic interactions for KS recognition. In contrast, the coulombic potentials are evenly distributed across the MAS ACP (Supplementary Fig. 17). This lack of electrostatic interactions is compensated by the multidomain architecture of MAS, where the ACP is located on the same polypeptide as other domains[31]. Interactions between ACP and KS are limited to a single hydrogen bond between Arg2045 on loop 1 of the ACP and Glu79 of KS (Fig. 3b and Supplementary Fig. 21a) along with weak hydrophobic and electrostatic interactions (Supplementary Fig. 50). Complex B reveals that the docking of ACP on KS is mainly facilitated by interactions with the neighboring AT domain via hydrogen bonding between Arg2032 and Arg2033 on helix 1 and Arg2090 on

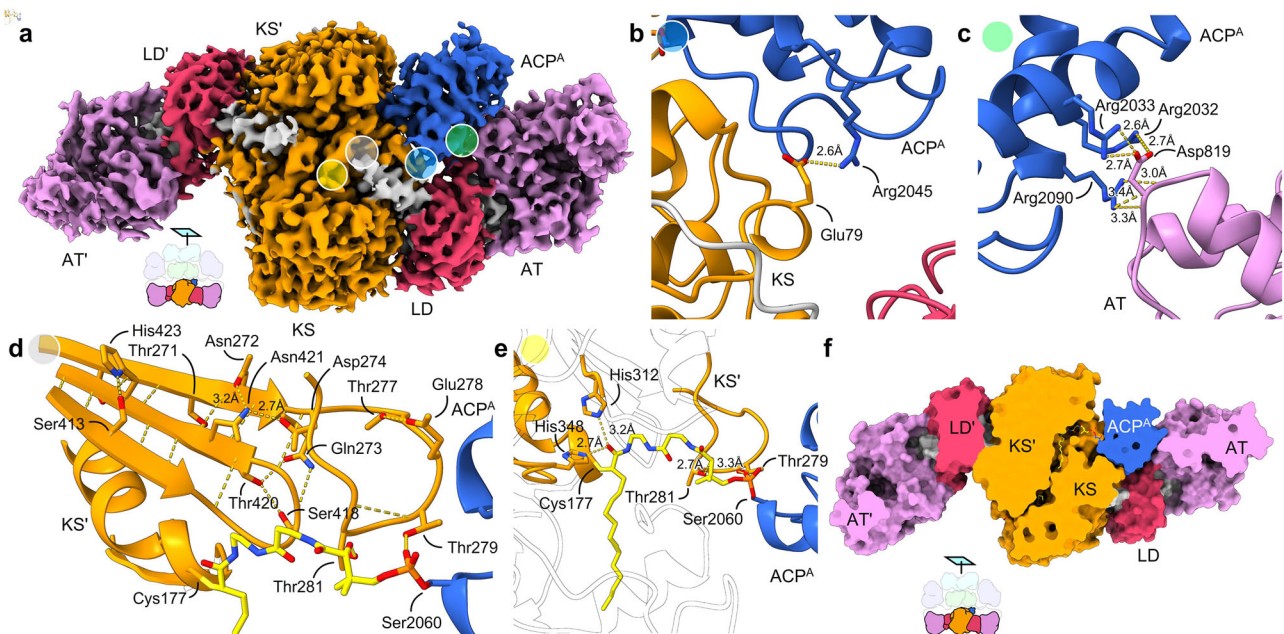

**Fig. 3 | Structure of complex B with a single ACP = KS crosslink. a** Map of one ACP crosslinked at KS' within the condensation compartment at 3.2 Å, with a contour level of 0.12. Colored circles represent regions of expansion in (**b**−**e**). Panel (**f**) further distinguishes the KS and KS'. **b** Crosslinking of ACP and KS traps PPIs between ACP and KS. A single hydrogen bond between Glu79 and Arg2045 mediates the ACP interaction with KS. **c** More extensive residue interactions are seen between ACP and AT. **d** The KS gating network (gating loops 1 and 2) residing at the entrance of the KS substrate binding tunnel demonstrates the inter-loop interactions (Asn272, Asp274, and Asn421) and its interactions with the PPant arm of **1**. This gate is visualized in both KSs in complex A. **e** Crosslinker **1** links ACP to KS by trapping Cys177 (Fig. 1b), highlighting active site residues in the substrate loading and reloading states (Fig. 1a). **f** Demonstration of the deep KS binding tunnel formed by the KS • KS' dimerization interface. In complex B, one ACP is crosslinked at KS, while KS is shunted by **1** without ACP.

helix 4 with Asp819 on the AT domain (Fig. 3C and Supplementary Fig. 21b).

We adopted a quantitative crosslinking assay to evaluate the MAS mutants in this study, as the correlation between crosslinking yield and enzymatic activity has been validated[32]. ACP-AT interactions were verified to be critical for the ACP catalytic efficiency with KS by evaluating comparative crosslinking of point mutants Glu79Lys, Asp819Ala, Asp819Arg, Arg2032Asp, and Arg2090Asp (Supplementary Figs. 22–27). In comparison, ACPs in modular type I PKSs DEBS-M1 (PDB ID: 7M7F, Supplementary Fig. 20b) and Lsd14 (PDB ID: 7S6C, Supplementary Fig. 20c) adopt slightly different binding modes at the condensation state, with more interactions between helix 1 and 2 of ACP and KS, but no interactions with the AT (Supplementary Fig. 20a)[13,14]. However, the observed ACP-AT interactions are in agreement with the recent complex between DEBS-M3 and ACP2 structure at the transacylation state[30]. It is worth noting that a six residue helix of KS near the ACP docking interface mediates interactions with ACP in Lsd14 (Pro118-Ser123, PDB ID: 7S6C, Supplementary Fig. 20c), DEBS-M1 (Pro113-Ala118, PDB ID: 7M7F, Supplementary Fig. 20b), as well as type II AcpP=KS (Ser63-Arg68, PDB ID:7L4L, Supplementary Fig. 20d); but this structural feature is not seen in MAS complex B (Supplementary Figs. 20a, e).

### Non-turnstile ketosynthase substrate regulation

The KS from MAS shares similarities in KS substrate binding with type I and II PKSs by using the same canonical KS binding tunnel as demonstrated in *E. coli* FabF (PDB ID: 7L4L), Lsd14 (PDB ID: 7S6C), and DEBS-M1 (PDB ID: 7M7F) (Supplementary Fig. 20e). The AcpP = FabF structure revealed an unprecedented gating motif for KSs that assists in the substrate binding sequence of the ping-pong mechanism[33]. Here, Phe400 of gating loop 1 moves 7 Å upon the interaction of gating loop 2 with *acyl*-AcpP in preparation for the initial transacylation step; this is followed by Phe400 repositioning near the active site for malonyl-AcpP to undergo decarboxylative Claisen condensation

(Supplementary Figs. 28b, c). In MAS complex B (Supplementary Fig. 28a), the crosslinker, tethered through a phosphodiester linkage to Ser2060 of ACP, has the PPant arm forming hydrogen bonds with residues Thr279 and Thr281 in KS (Fig. 3d, e). These reside on a loop analogous to gating loop 2 of FabF, which orchestrates with gating loop 1 to govern the recognition and processing of ACP substrates (Supplementary Figs. 28b, c)[33,34]. In *E. coli* FabF, PPant interactive residues His268 and Thr270 in gating loop 2 are responsible for governing the gate-closed and gate-open conformation by interacting with Asp35 of AcpP. In KS, Thr279 and Thr281, respectively, reflect similar positions (Supplementary Fig. 28).

Through the crosslinking assay (Supplementary Fig. 22), mutant Thr279Ala (Supplementary Figs. 22–23) revealed a 2-fold decrease, and Thr281Ala (Supplementary Figs. 22–23) showed complete abolishment of crosslinking efficiency. Based on this result, Thr279 and Thr281 in gating loop 2 are strong candidates for regulating the gating mechanism of KS. Consistent with FabF, the coordination of gating loops 1 and 2 is directed by the highly conserved residues Asp274 and Asn421 across type I and type II PKS and FAS KSs (also conserved in DEBS-M1 and LSD14)[33]. Surprisingly, crosslinking was completely eliminated in MAS mutant Asn421Ala, suggesting its potential role in orchestrating the KS gating system (Supplementary Fig. 22). Although the gating loops are in the closed conformation upon crosslinking in both MAS complexes A and B, Asp2059 of ACP appears to be a candidate residue to trigger opening of the gating network by interaction with loop 1 of KS (Supplementary Fig. 28a). Interestingly, the main KS gating residue (gating loop 1) is Met417 instead of Phe, as found in type II PKS and FAS systems[33,34]. When mutated to Phe (Met417Phe), a 35% decrease in crosslinking efficiency was observed. This may be due to the use of methylmalonyl-CoA as the extension unit in MAS instead of malonyl-CoA, as Met is more flexible and less bulky. Although more detailed investigation is necessary, key residues Thr279, Thr281, Met417, and Asn421 are most probably involved in regulating the

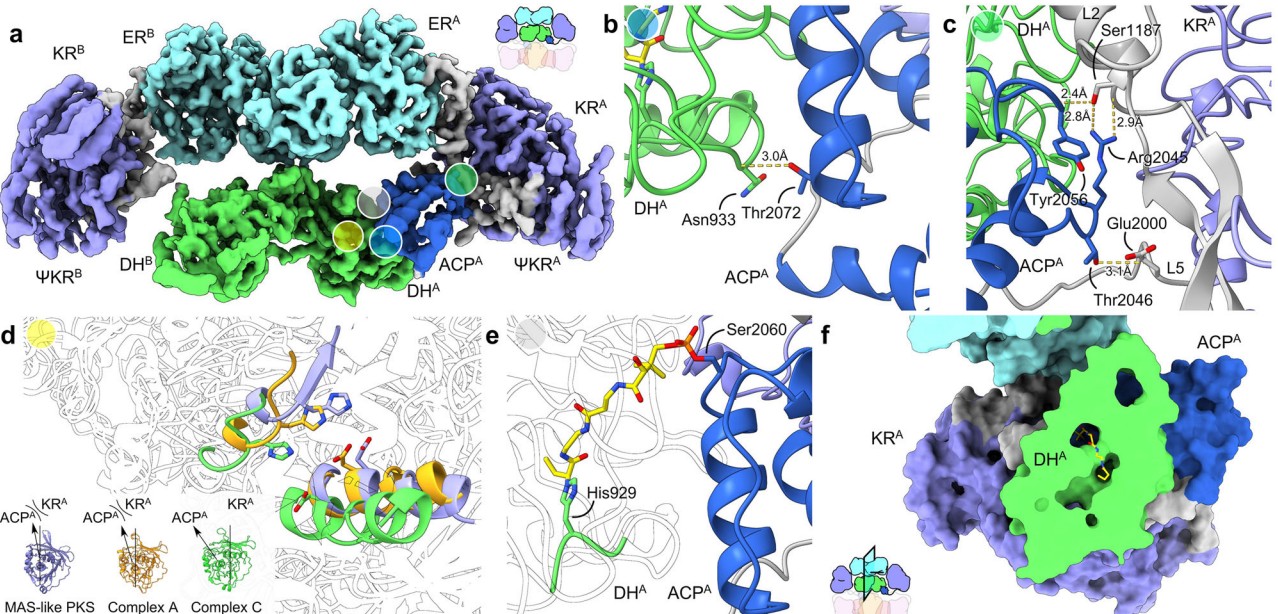

**Fig. 4 | Structure of single ACP = DH crosslinked complex C. a** Cryo-EM map at 3.7 Å and a contour level of 0.19, of ACP crosslinked to DH of the modifying compartment. Colored circles represent regions of expansion in (**b**–**e**). **b** Upon ACP = DH crosslinking, interactions are observed between ACP with DH. A single hydrogen bond is observed between DH Asn933 and ACP Thr2072. **c** In addition, more extensive interactions are seen between ACP and L2 as well as L5. **d** Comparison amongst *apo*-DH of MAS-like PKS (purple) (PDB ID: 5BP4), *apo*-DH of MAS complex A (orange) (PDB ID: 9D2Y), and crosslinked ACP = DH of complex C (PDB ID: 9D30) demonstrating DH reactive residues His and Asp as well as the significant domain translation and rotation (Supplementary Fig. 39). DH domains of MAS-like PKS and Complex A were aligned to the crosslinked DH of complex C (RMSD = 1.266 Å). Residues shown are the reactive His and Asp residing at the active site of the DH domain. Lower left cartoon demonstrates the crosslinked DH substrate binding tunnel entrance (arrow) translates and rotates away from KR to be accessible for MAS ACP crosslinking in complex C. **e** ACP = DH crosslinking forged a covalent bond between active site His929 and **1**. The continuous map density for the crosslinking at His929 and Ser2060 can be found in Supplementary Fig. 36. **f** Depiction of the hydrophobic tail of **1** within the binding pocket of DH (Supplementary Fig. 37).

gate-open and gate-closed conformations in KS as a part of the substrate regulation.

Within the KS active site, the covalent bond between the α carbon of **1** and Cys177 of KS is well visualized (Supplementary Fig. 18). The warhead carbonyl of crosslinker **1** coordinates with two highly conserved residues His312 and His348 (Supplementary Fig. 18), trapping the transacylation (substrate loading or reloading) state of KS in its gate-closed form (Supplementary Fig. 28a)[19]. Supported by continuous map density, the C16 acyl tail of **1** is buried deep within the hydrophobic tunnel formed by the KS • KS' dimeric interface (Supplementary Figs. 18 and 29). The end of the binding tunnel lacks a clear boundary with the binding tunnel of the other KS monomer, providing space for the final long, branched mycocerosic acid product (Supplementary Fig. 29). In comparison to KS from DEBS-M1 (Supplementary Fig. 29c) and Lsd14 (Supplementary Fig. 29d), the binding tunnels of MAS and FabB KSs (Supplementary Fig. 29b) are narrower. A published MAS-like PKS KS-LD-AT tridomain structure (PDB ID: 5BP1) was crystalized in the monomeric form, and the KS domain was reported to be catalytically inactive[5]. Compared with the intact MAS complex B, active site Cys178 is shifted by 10.2 Å. The dimerization interface in complex B (Supplementary Fig. 30) plays a pivotal role in maintaining the stability and positioning of the active site. In the other protomer's KS domain, **1** can be found alone (a shunt product without ACP) with continuous density. We presume that the highly potent warhead reacted with the KS active site as a small molecule in the absence of ACP[35].

The AT domain in complex A is in the *apo* form (without substrate in the active site), consisting of an α/β-hydrolase and ferredoxin subdomains, sharing a similar fold with other type I and type II systems. The "turnstile closed" conformation shown in the reported DEBS-M1 is not observed in either complex A or B, leaving the KS domain accessible to ACPs[13,36]. The turnstile model in type I modular PKS proposed

by Khosla et al. emphasizes the reloading of extended or processed substrate is prohibited by the "turnstile closed" conformation (AT blocking ACP access to the KS) to ensure the vectorial synthesis of modular polyketides[37]. The absence of observed "turnstile closed" conformations in MAS complexes A and B implies that iterative type I PKSs utilize a different mechanism to prevent premature reloading of the KS domain.

**Intramolecular crosslinking between DH and ACP in Complex C**

We also obtained a map of the modifying compartment DH-ΨKR-ER-KR-ACP (3.74 Å) from the same dataset. Here, one ACP is covalently crosslinked to the DH domain (Fig. 4a and Supplementary Figs. 31–33), which consists of a double-hotdog fold in type I PKSs. In this complex, **1** covalently crosslinks the ACP to the active site His929 of the DH (Fig. 4e and Supplementary Fig. 36). Like the observations found with complexes A (Supplementary Fig. 10) and B (Supplementary Fig. 15), local resolution on DH^A is higher compared to that of the uncrosslinked DH^B (Supplementary Fig. 31). Even though DH is composed of a double-hotdog fold, it only contains one set of reactive residues His797 and Asp1095 (Supplementary Fig. 34a, c). The other hotdog fold is responsible for forming the DH^A • DH^B dimerization interface through a "handshake fold"[5]. The two hotdog folds are connected through a 20-residue loop (Supplementary Fig. 34b). The ACP = DH crosslinking observed in complex C is bound in an opposite orientation when compared to that for the *E. coli* AcpP=FabA (PDB ID: 4KEH) (Supplementary Fig. 34a)[38]. In contrast to the extensive interactions between the highly negatively charged AcpP (helices 2 and 3) and a positively charged patch of FabA, the ACP in MAS shows only limited interactions with the DH domain between Thr2072 of helix 2 of ACP and Asn933 of DH (Fig. 4b and Supplementary Figs. 33 and 35a). Akin to the binding between ACP and KS in complexes A and B, the binding

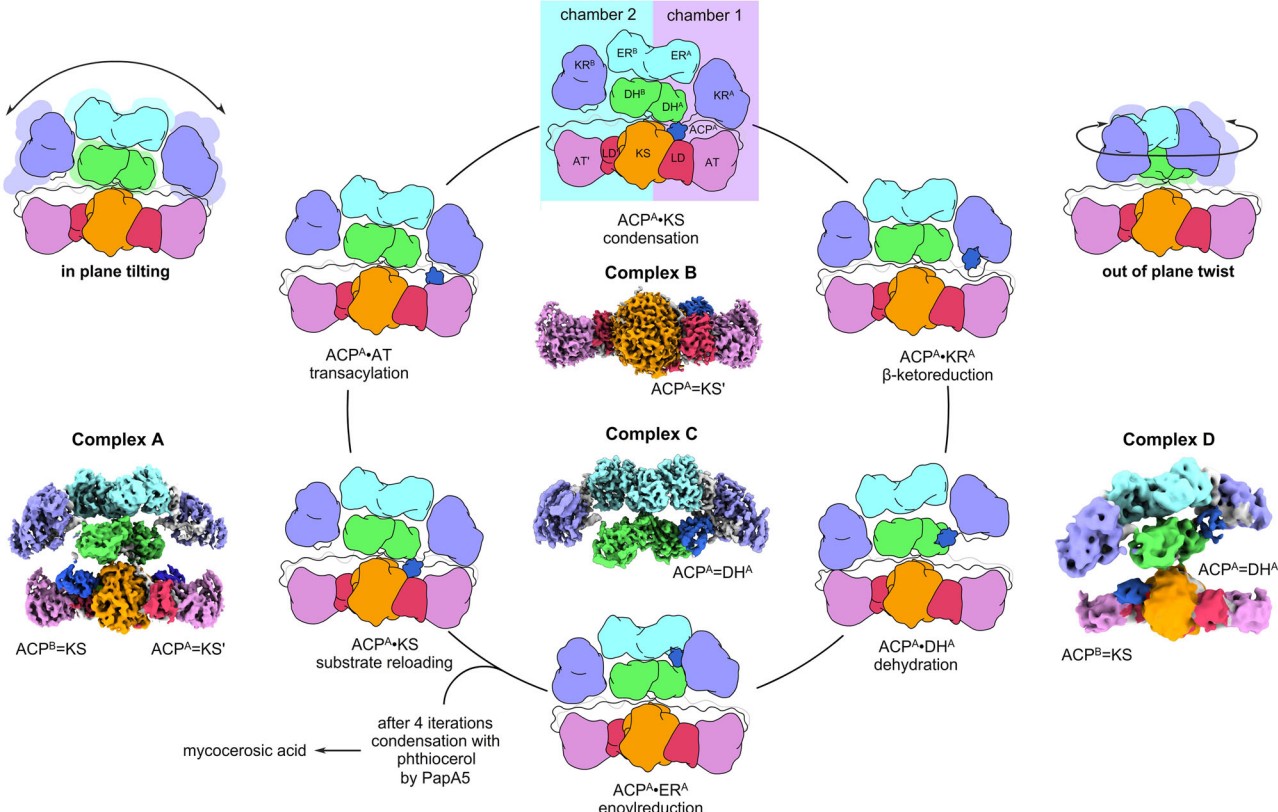

**Fig. 5 | Structural insight into MAS conformationally-guided processivity.**
Following KS (at KS and KS'), ACP•KS substrate reloading occurs through transacylation from ACP. Next, ACP accepts a methylmalonate monomer from AT. ACP shuttles the extension unit to its partner KS to induce condensation. Complex B informs this step. MAS flexibility enables in plane tilting and out-of-plane twisting. During this process, ACP is distal to the condensation compartment, therefore, interactions with the modifying compartment are favored, and the ACP binds

sequentially to KR, DH, and ER. Dehydration processing is exemplified in complexes C and D, where complex D is trapped through crosslinking of ACPs with DH (chamber 1) and KS (chamber 2), respectively, showing this conformational movement. A full iteration is completed through substrate reloading. After 4 iterations, PapA5 transfers the mature mycocerosates to phthiocerol, forming phthiocerol dimycocerosate (PDIM).

between ACP and DH is mainly facilitated by surrounding structural elements. Arg2045, Thr2046, and Tyr2056 (ACP loop 1) interact with Ser1187 and Glu2000 of inter-domain linkers (L2 and L5) (Fig. 4c and Supplementary Fig. 35b). Mutation of Ser1187Ala and Thr2046Val validated these interactions as important for activity, resulting in a loss of 22% and 20% crosslinking, respectively. These findings emphasize the outsized role that inter-domain linkers play in PKS catalysis.

Although crosslinker **1** has historically been used to target KS, complex C only contains crosslinking between ACP and DH. Unlike the shunt product in complex B (Fig. 3f), there is no probe observed in the uncrosslinked DH of the opposite protomer, indicating that the reactivity between crosslinker **1** and DH is facilitated by ACP binding. To validate the ability of α-bromoamide warheads to crosslink domains other than KS, we performed crosslinking experiments of *E. coli crypto*-AcpP loaded with **2, 3, 5**, or **6** with a panel of partner proteins including FabF (KS), Cys163Ala FabF (KS*), FabG (KR), FabA (DH), FabI (ER), and FabD (AT) from *E. coli* (Supplementary Fig. 6). Similar to our observation in MAS complex C, where ACP is crosslinked with DH, *crypto*-AcpPs loaded with **2** and **3** were able to crosslink both FabA and FabF, with no crosslinking seen with the other enzymes.

Crosslinker **1** enters the canonical DH substrate binding tunnel and is crosslinked to His929 (Fig. 4e and Supplementary Fig. 37A), similar to the type II *E. coli* AcpP=FabA (PDB ID: 4KEH, Supplementary Fig. 37b) and AcpP=FabZ (PDB ID: 6N3P) structures[38,39]. Complete map density is observed from Ser2060 to C4 of the acyl tail of **1**, although no other significant substrate-protein polar interactions are observed (Supplementary Fig. 36). Unlike the narrow and well-defined FabA

active site, MAS-DH features a deep binding tunnel to accommodate the long, branched-chain acyl intermediates (Fig. 4f and Supplementary Fig. 37). The analogous *E. coli* crosslinked structure (AcpP = FabA) revealed a DH gating mechanism utilizing residues Phe165 and Phe171 to enforce substrate specificity (Supplementary Fig. 38b). In MAS-DH, these gating residues are replaced by the DH-ΨKR interdomain linker L2, leaving the DH entrance exposed (Supplementary Fig. 38a, c). Surprisingly, while the *E. coli* type II DH regulates the substrate specificity via gating residues, we observe a major domain rotation of the crosslinked DH in comparison to the *apo*-DHs in MAS complex A and MAS-like PKS modifying compartment (Fig. 4d and Supplementary Fig. 39). In the DH *apo* form the active site entrance resides underneath the ΨKR • KR domains, providing no access for ACP binding. In complex C, the DH translates and rotates away from the ΨKR • KR to enable access to the ACP (Supplementary Fig. 39). The limited space between *apo*-DH and ΨKR • KR hinders ACP docking and likely plays a role in shielding the active site of the DH in certain catalytic states.

## ACP = DH and ACP = KS Dual Crosslinked Complex D
Within the same data set, a 6.18 Å density map was obtained visualizing both ACPs trapped at DH and KS, respectively, yielding complex D (Fig. 5 and Supplementary Fig. 40a). While this reconstruction was not detailed enough to allow atomic modeling, it reveals the remarkable flexibility of MAS processivity. This map shows an asymmetric pose of MAS, and all domains are visualized, including both ACPs. Anchoring at the KS • KS' core, the modifying compartment DH^B-ΨKR^B-ER^B-KR^B demonstrates an astonishing higher-order dynamic (tilted and twisted)

pose directed toward KS • KS', where ACP$^B$ is trapped at KS (Supplementary Fig. 40b). On the other side, DH$^A$-ΨKR$^A$-ER$^A$-KR$^A$ is tilted and twisted away from the KS • KS' core, with ACP$^A$ trapped at DH. In contrast to complex A, where both ACPs are crosslinked at KSs, showing only a slight tilt, the conformational state of complex D clearly revealed the capability of catalytic chambers 1 and 2 (Fig. 2a) to process substrates asymmetrically. The fact that both ACPs are trapped at different domains in two different catalytic chambers in complex D, together with complex A, in which both ACPs are crosslinked at KSs, reveals that both chambers can process substrates synchronously, in contrast to DEBS-M1 and Lsd14 type I modular systems[13,14]. At maximum extension, the length between active site residues Ser2060 (ACP$^A$) and Ser1996 (L5) is 84 Å (Supplementary Fig. 41). The distance between Ser1995 (KR$^A$) to the phosphodiester motif of the PPant in KS (chamber 1, Fig. 2a) is approximately 66 Å. When DH-ΨKR-ER-KR is tilted and twisted away from the KS • KS' core, ACP$^A$ active residue Ser2060 can still reach the KS • KS' core through the length of L5; however, the modifying domains KR$^A$, DH$^A$, and ER$^A$ are preferred. In the comparison amongst the linker conformations of complexes A, B, and C, the tilting and twisting movements stem from linker 1 (unmodeled residues 882-893 at the C-terminal). Linkers 2, 3, 4, and most of linker 1 adopt similar conformations amongst complexes A, B, and C. Linker 5, which directly governs the flexibility of ACP, on the other hand, points in different directions between complexes A and C. (Supplementary Fig. 52) The tilting and twisting conformation demonstrates that type I PKSs can make use of conformational flexibility to localize reactivity between the distinct elongation and modification compartments, much like what has been indicated with type I FAS[40].

## Discussion

The combination of developed dual covalent crosslinkers and the ability of cryo-EM to capture multiple protein complexes within a single dataset has enabled the visualization of multiple trapped catalytic states in the type I iterative PKS MAS from *M. tuberculosis*. Crosslinking between ACP and catalytic domains led to an overall rigidified protein architecture, provided resolution improvement, and yielded four uniquely trapped states, complexes A–D. Upon crosslinking, the ACP = KS and ACP = DH binding interfaces were revealed, highlighting interactions between these domains as well as with neighboring structural elements (Supplementary Fig. 42). In comparison to type II systems, the ACP in MAS adopted unique docking orientations at both KS and DH. Importantly, the expected PPIs between ACP and each of these domains were altered and largely replaced by PPIs with adjacent domains and linker peptides. In addition, crosslinker **1** interrogated the substrate-protein interactions in both the KS and DH domain active sites. The inability to crosslink truncated KS or KS-AT with ACP emphasizes the critical role of domain architecture and linker regions in stabilizing reactive interactions. Importantly, complexes A and B are the validation of cross-protomer processivity in type I PKSs based on the SDS-PAGE results (Supplementary Figs. 7 and 45), while crosslinking within the same protomer is likely due to high out-of-plane twisting. Complex D shows that while ACP interacts with DH in chamber 1, the other ACP interacts with the KS in chamber 2 (Fig. 5), corroborating the obligate dimerization of MAS. The cross-protomer processivity plays an important role in guiding synthetic efficiency in type I PKSs, and likely FASs by allowing the ACP$^A$ to interact with KS while ACP$^B$ interacts with KS' or vice versa.

In complex B, MAS potentially adopts KS gating loops similar to *E. coli* type II KSs, and the presence of key gating residues is essential for activity. In complex C, MAS-DH does not demonstrate gating control like that found in *E. coli* FabA and *H. pylori* FabZ[39,41–43]. Instead, DH rotation upon ACP binding presents a strong candidate for DH gating regulation. Based on the structural features of MAS observed in all complexes, an iterative assembly line mechanism can be suggested

(Fig. 5). Starting with the substrate loading step, *acyl*-ACP$^A$ interacts with KS in catalytic chamber 1 to deliver a long fatty acyl thioester to Cys177. Based on the higher-order dynamics in this step[40], if in-plane tilt and out-of-plane twist are moderate, ACP$^B$ can interact with the KS in the other chamber, exemplified by complex A. If the modifying compartment (DH$^B$-ΨKR$^B$-ER$^B$-KR$^B$) in chamber 2 tilts and twists away from the KS domain, ACP$^B$ kinetically prefers to interact with the modifying compartment, supported by complexes B and D. This high degree of twist and tilt may be the reason that the KS domain is inhibited by shunt product in complex B.

Subsequently, *holo*-ACP visits the AT to load a methylmalonyl extender unit, followed by a return to the KS for chain elongation in chamber 1 (Fig. 5). These three steps, substrate loading, transacylation, and condensation, require ACP to shuttle within the condensation compartment in chamber 1. As the modifying compartment tilts and twists away from the condensation compartment in chamber 1, the interactions of ACP$^A$ with modifying domains in chamber 1 are more favorable. In the reduction steps, the elongated substrate on ACP is fully reduced in the modifying compartment in chamber 1 (Fig. 5). The dehydration step is structurally represented by complexes C and D. Complex D demonstrates ACP trapped at KS in one chamber, while ACP is trapped at DH in the other. After the substrate on ACP is fully reduced, it is reloaded onto KS for another iteration. Such a higher-order dynamics-driven vectorial processivity agrees with the "see-saw" model suggested by EcPKS1 and EcPKS2 from *Elysia chlorotica* by the Hill and Schmidt groups[44]. In the EcPKS2 structure, the linker between KR and ACP alternates between disordered and ordered conformation (interacting with nonfunctional methyltransferase) to control the flexibility of ACP and, consequently, change the domain sampling profiles. Although the alternation between ordered and disordered conformations of linker 5 was not found in the MAS complexes, our model does not exclude this function.

In modular PKS Lsd14, the post-ACP dimerization element or pre-KR dimerization element keeps both ACPs in proximity[14,45,46]. Due to this, the catalytic chambers function asynchronously in their proposed pendulum model. The absence of post-ACP dimerization element or pre-KR dimerization element allows MAS to function synchronously and asymmetrically (Supplementary Fig. 51). In DEBS-M1, to strictly enforce the vectorial biosynthesis, the protein adopts a turnstile-closed conformation after chain elongation, where the AT interacts with the LD to sterically occlude the ACP from binding KS. These structural features in modular PKS systems prevent substrate reloading onto KS to avoid unwanted products. However, in iterative type I PKS systems such as MAS, substrate reloading is essential for processivity. The higher-order dynamics can guide the ACP to interact preferentially with the condensation or modifying compartments, but it does not completely prohibit ACP from revisiting domains. Therefore, a substrate regulation mechanism is needed in iterative type I PKS system. In the MAS KS, while gating loops 1 and 2 may play a role in substrate specificity, recent studies have shown the substrate gatekeeping capability of KS in type I system[31,47,48]. Mutagenesis and analysis depicted that KS dimer interface loops containing conserved TNGQ or VMYH motifs are commonly found in KS families that preferentially accept D-β-hydroxyacyl or α/β-enoylacyl intermediates, respectively. Such a dimer interface loop is not conserved in KSs that accept β-methylene intermediates. MAS KS, which does not contain the TNGQ or VMYH motifs, instead possesses LTHE (residues 127-130 in MAS complex B), hinting that D-β-hydroxyacyl or α/β-enoylacyl intermediates are not the preferred substrates. KS substrate specificity in a type I system was also observed in our recent development of cerulenin crosslinker **8** (Supplementary Fig. 3)[20]. Due to the oxidized α, β, and γ positions in the cerulenin crosslinker, it had minimal crosslinking between MAS ACP and MAS KS. The fact that crosslinking between MAS ACP and MAS KS can be achieved by fully reduced crosslinker **1** also indicates the fully reduced substrate preference of

MAS-KS. In terms of substrate chain length, it is noteworthy that the two substrate tunnels formed by the KS • KS' dimer in complex B are connected (Fig. 2f and Supplementary Fig. 29a), contributing to a potential substrate tunnel sharing to accommodate C22-C28 mycocerosic acids products. In contrast, FabB does not have shared tunnels. Its substrate chain length is controlled by a "back gate" consisting of two sets of Glu200 and Gln 113. The coordination of the "back gate" controls the size of the tunnels asymmetrically[49]. This negative cooperativity contributes to the acceptance of long substrates in one tunnel, while the other tunnel is limited to accept short substrates (Supplementary Fig. 43). After the condensation step at the KS domain, the elongated substrate does not visit the AT domain, as the AT, although proximal, strictly accepts methylmalonyl-CoA[50]. This substrate-based control is different from type II ACPs, which sequester acyl substrates, thereby altering the ACP binding interface and achieving preferential binding with specific partner proteins through allostery[51]. Since type I ACPs do not sequester their acyl substrates, the interactions with different domains are more likely guided by the loaded acyl substrates, as proposed for the pikromycin PKS[52].

Here we show how a dual covalent crosslinker in concert with cryo-EM can reveal previously inaccessible structural features of type I PKSs. The crosslinked species can be reliably resolved at high resolution due to constraints of dynamic movement in crosslinked megasynthase compartments. The ability for a single crosslinker to interact with more than one domain allows the collection of multiple complexes within a single data set that can be differentiated with particle sorting and reconstruction. The versatility of this method parallels prior findings in type II synthases solved by X-ray crystallography, hereby revealing the elegance of the type I PKS organization and control. Compared with non-specific crosslinkers, dual covalent crosslinkers offer not only the visualization of PPIs, but also substrate-protein interactions. Most importantly, covalent crosslinkers facilitate structural rigidification, which gives rise to higher resolution density maps. Most importantly, the dual site-selective crosslinking approach used herein reveals structures of states actively trapped during their enzymatic process. Recent studies, including those on human FAS[53,54] show that one can apply cryo-EM with high data content to refine structures resting at specific positions within the catalytic process, including reduction at the ER, dehydration at the DH, condensation at the KS, and acyl transfer at the AT. The fact that these states are resting means that they do not necessarily provide accurate representations of substrate undergoing enzymatic processing, but dual site-selective state trapping, as well as other approaches to enzymatically lock states during their reactions do.

Using this technology, we reveal four informative MAS complexes from a single data set, visualizing both ACPs from the MAS homodimer in action. We show the ACP binding interfaces with KS and DH, interactions with adjacent domains and linkers, and substrate-protein interactions. While the KS of MAS potentially adopts a gating mechanism similar to type II KSs, DH rotation upon ACP crosslinking was identified as a potential DH gating mechanism. Complexes A and D suggest that both catalytic chambers can not only be functional synchronously but also asymmetrically, constructing a model that allows the ACP trajectory to follow the sequential catalytic loop (transacylation, condensation, reduction, and substrate reloading) controlled by substrate selectivity. The cross-protomer activity of ACP-KS interactions provides a model for multidomain catalysis. Altogether, this study provides fundamental concepts and structural details of a mycocerosic acid-producing machinery that offers potential as a therapeutic target.

## Methods
### General experimental methods
For protein expression, LB media (RPI Research Products), agar (RPI Research Products), IPTG (Bio Pioneer), ampicillin (Fisher Scientific),

and kanamycin (Apex BioResearch Products) were used. For buffer preparation, Tris base (Fisher Scientific), HCl (Fisher Scientific), NaCl (RPI Research Products), glycerol (Honeywell Research Chemicals), imidazole (MP Biomedicals), maltose (Sigma Aldrich), $MgCl_2$ (BDH Chemicals), $Na_2HPO_4$ (Fisher Scientific), $KH_2PO_4$ (Fisher Scientific), KCl (Fisher Scientific), ATP (Thermos Scientific), DTT (Fisher Scientific), and DMSO (Fisher Scientific) were used. For protein purification, we used a sonicator (Ultrasonic Processor FS-600N), Ni-NTA columns (Thermo Scientific), amylose columns (New England BioLabs), 10 K, 30 K, 50 K, and 100 K MWCO spin concentrators (Amicon Millipore), 3 K and 10 K MWCO Snake Skin dialysis tubing (Thermo Fisher), Hiload Superdex 200 pg (16 × 600) (GE Biosciences), HiLoad Superdex 75 pg (16 × 600) (GE Biosciences) size exclusion columns, and an FPLC (ÄKTA Pure). In reactions that contain crosslinkers 1–12 and fluorescent probe 13 (dissolved in DMSO), the DMSO concentration was kept below 1% in the final reaction. The preparation of 1-13 was prepared using previously published protocols[20–22,55,56]. Materials were used with a purity >98% by NMR. Typically, stock solutions in DMSO were prepared fresh and used as is. For gel preparation or evaluation, SDS (Sigma Aldrich), 40% acrylamide/bis-Acrylamide 29:1 (Fisher Scientific), TEMED (Sigma Aldrich), APS (Fisher Scientific), Wide Range Gel Preparation Buffer (4×) (Nacalai Tesque), Mini-Protean TGX gel (4–15%) (Bio-Rad), and 4–12% bis-Tris NuPage (Invitrogen) were used. For gel imaging, UV transilluminator at 365 nm was used prior to Coomassie Brilliant Blue staining. For gel-based crosslinking yield quantification, Perfection V19 scanner (Epson) and ImageJ[57] were used. When possible raw images were used to make the figures. Unedited gel images were used or provided in Supplementary Fig. 46.

### Cloning of M. tuberculosis MAS and variants
Primers for cloning were ordered from Integrated DNA Technologies. Restriction enzymes, Gibson master mix, and T4 ligase were ordered from New England Biolabs. Sanger sequencing was performed by Azenta Life Sciences. Wild-type MAS from *M. tuberculosis* H37Rv (UniProt ID: I6Y231) in a pET21 vector (pET23) was provided by H. Gramajo[58]. Excised KS from *M. tuberculosis* MAS was synthesized in pET28 as a codon-optimized construct by Twist Bioscience. The boundaries of *M. tuberculosis* KS-LD-AT were determined through sequence alignment with MAS-like type I PKS from *M. smegmatis* (UniProt ID: A0R1E8). Excised KS-LD-AT tridomain was subcloned from pET23 into pET21 using Gibson cloning (NEB HiFi assembly) and directly transformed into DH5α cells. Resulting colonies were grown overnight in LB broth (5 mL) with ampicillin, miniprepped, and sent for Sanger sequencing to confirm proper insertion. The ACP of *M. tuberculosis* MAS was synthesized by Twist Bioscience in a pET21 expression vector.

### Expression and purification of M. tuberculosis MAS
C-terminal His$_6$-tagged recombinant MAS was expressed in *E. coli* BL21(DE3) similar to that previously described in refs. 20,21. A single colony with pET23 was cultured in LB broth (5 mL) with 100 mg/L of ampicillin for 12 h at 37 °C. This culture was added into LB broth (1 L) with 100 mg/L of ampicillin and incubated with shaking at 37 °C until the $OD_{600}$ reached 0.6–0.8. The culture was then cooled on ice followed by induction with 0.5 mM IPTG (addition of 1 mL of 500 mM IPTG) at 37 °C and left shaking for 18 h. Cell pellets were obtained through centrifugation (4816 × g for 30 min). The cell pellets were resuspended in 25 mL cold lysis buffer (50 mM Tris • HCl pH 7.4, 150 mM NaCl, 10% glycerol), submerged in ice, and lysed by sonication (1 s on and 3 s off cycle for 9 min). The lysate was then centrifuged at 4816 × g for 1 h. The resulting supernatant was loaded onto Ni-NTA column (Thermo Fisher Scientific) and washed with 10 CV (column volume) of the lysis buffer followed by lysis buffer containing 10 mM imidazole (2 × 10 mL). Lysis buffer containing 250 mM imidazole was used to elute the protein. Dialysis with 10 K MWCO SnakeSkin tubing

(Thermo Fisher Scientific) was performed with lysis buffer (1 L) for 18 h at 4 °C. The dialyzed protein was then subjected to HiLoad Superdex 200 pg (16 × 600) (GE Biosciences) size exclusion column purified with lysis buffer (50 mM Tris • HCl pH 7.4, 150 mM NaCl, 10% glycerol). Fractions containing pure MAS were collected and concentrated to 32 mg/mL using a 100k Amicon Ultra spin concentrator (Millipore) and flash frozen in aliquots for storage at −80 °C.

## Expression and purification of excised KS and excised KS-LD-AT

The expression and purification of excised KS and excised KS-LD-AT follow a common protocol. C-terminal His6-tagged recombinant excised KS and excised KS-LD-AT[20] were expressed in BL21(DE3) and cultured in LB broth (5 mL) with 100 mg/L of ampicillin for 12 h at 37 °C. The culture was added into LB broth (1 L) with 100 mg/L of ampicillin and incubated at 37 °C until the OD600 reached 0.6–0.8. The culture was then cooled on ice, followed by induction with 0.5 mM IPTG (addition of 1 mL of 500 mM IPTG) at 37 °C and left shaking for 18 h. Cell pellets were obtained through centrifugation (4816 × g for 30 min). The cell pellets were resuspended in 25 mL of cold lysis buffer (50 mM Tris • HCl pH 7.4, 150 mM NaCl, 10% glycerol), submerged in ice, and lysed by sonication (1 s on and 3 s off cycle for 9 min). The lysate was then centrifuged at 4816 × g for 1 h. The resulting supernatant was loaded onto a Ni-NTA column (0.5 mL/L of culture) (Thermo Fisher Scientific) and washed with 10 CV of the lysis buffer, followed by lysis buffer containing 10 mM imidazole (2 × 10 mL). Lysis buffer containing 250 mM imidazole was used to elute the protein. Dialysis with 10 K MWCO SnakeSkin tubing (Thermo Fisher Scientific) was performed with lysis buffer (1 L) for 18 h at 4 °C. The dialyzed protein was then subjected to HiLoad Superdex 200 pg (16 × 600) (GE Biosciences) size exclusion column and purified with lysis buffer (50 mM Tris • HCl pH 7.4, 150 mM NaCl, 10% glycerol). Fractions containing excised KS or excised KS-LD-AT were collected and concentrated to 32 mg/mL using a 10k Amicon Ultra spin concentrator (Millipore) and flash frozen in aliquots for storage at −80 °C.

## Expression and purification of excised ACP from MAS

The expression and purification of excised ACP adopted a reported protocol[59]. C-terminal His6-tagged recombinant excised ACP was expressed in BL21(DE3) and cultured in LB broth (5 mL) with 100 mg/L of ampicillin for 12 h at 37 °C. The culture was added into LB broth (1 L) with 100 mg/L of ampicillin and incubated with shaking at 37 °C until the OD600 reached 0.6–0.8. The culture was then cooled on ice followed by induction with 0.5 mM IPTG (addition of 1 mL of 500 mM IPTG) at 37 °C and left shaking for 18 h. Cell pellets were obtained through centrifugation (4816 × g for 30 min). The cell pellets were resuspended in 25 mL of cool lysis buffer (8 mM Na2HPO4, 286 mM NaCl, 1.4 mM KH2PO4, 2.6 mM KCl, and 1% SDS (w/v) pH 7.4), submerged in ice, and lysed by sonication until the lysate turns clear (1 s on and 3 s off sonication cycle). The sonicated lysate was then chilled on ice for 1 h to cause SDS precipitation. To remove the SDS precipitates, the cold lysate was centrifuged at 4816 × g for 30 min at 4 °C. The resulting supernatant was loaded onto a Ni-NTA-column (0.5 mL/L of culture) (Thermo Fisher Scientific) and washed with 10 CV of the washing buffer (8 mM Na2HPO4, 286 mM NaCl, 1.4 mM KH2PO4, 2.6 mM KCl, 0.15% tween 20 (w/v), and 15 mM imidazole, pH 7.4) followed by elution buffer (1 mL × 7) (8 mM Na2HPO4, 286 mM NaCl, 1.4 mM KH2PO4, 2.6 mM KCl, 0.15% tween 20 (w/v), and 500 mM imidazole, pH 7.4). Dialysis with 3 K MWCO SnakeSkin tubing (Thermo Fisher Scientific) was performed with dialysis buffer (1 L) (8 mM Na2HPO4, 286 mM NaCl, 1.4 mM KH2PO4, 2.6 mM KCl, and 10% glycerol) for 18 h at 4 °C. The dialyzed protein was collected and concentrated to 3 mg/mL using a 3k Amicon Ultra spin concentrator (Millipore) and flash frozen in aliquots for storage at −80 °C.

## MAS mutant site-directed mutagenesis

MAS mutants were made using Quikchange Mutagenesis. Primers were ordered from Integrated DNA Technologies. Q5 Hot Start High Fidelity polymerase and restriction enzyme DpnI were ordered from New England Biolabs. Whole-plasmid sequencing was performed by Primordium Labs. Briefly, PCR was performed for 35 cycles, followed by template digestion with DpnI at 37 °C for 3 h and inactivation at 80 °C for 20 min. Digested samples were directly transformed into 10-beta chemically competent cells and plated at 37 °C overnight with the corresponding selection antibiotics. Colonies were screened the following day by whole-plasmid sequencing to validate the mutation.

## Expression and purification of MAS mutants

MAS mutants Asp819Arg, Asp2050Arg, Arg2090Asp, Glu79Lys, Asp819Ala, Arg2032Asp, Thr281Ala, Thr279Ala, Met417Phe, Asn421Ala, Ser1187Aal, Thr2046Val, Cys177Ala, and His929Ala were prepared by a common protocol. C-terminal His6-tagged recombinant MAS mutants[20] were expressed in BL21(DE3) cells and cultured in LB broth (5 mL) with 100 mg/L of ampicillin for 12 h at 37 °C. The culture was added into LB broth (1 L) with 100 mg/L of ampicillin and incubated with shaking at 37 °C until the OD600 reached 0.6–0.8. The culture was then cooled on ice, followed by induction with 0.5 mM IPTG (addition of 1 mL of 500 mM IPTG) at 18 °C and left shaking for 18 h. Cell pellets were obtained through centrifugation (4816 × g for 30 min). The cell pellets were resuspended in 25 mL of cold lysis buffer (50 mM Tris • HCl pH 7.4, 150 mM NaCl, 10% glycerol), submerged in ice, and lysed by sonication (1 s on and 3 s off cycle for 9 min). The lysate was then centrifuged at 4816 × g for 1 h. The resulting supernatant was loaded onto a Ni-NTA column (0.5 mL/L of culture) (Thermo Fisher Scientific) and washed with 10 CV of the lysis buffer, followed by lysis buffer containing 10 mM imidazole (2 × 10 mL). Lysis buffer containing 250 mM imidazole was used to elute the protein. Dialysis with 10 K MWCO SnakeSkin tubing (Thermo Fisher Scientific) was performed with lysis buffer (1 L) for 18 h at 4 °C. The dialyzed protein was then subjected to HiLoad Superdex 200 pg (16 × 600) (GE Biosciences) size exclusion column and purified with lysis buffer (50 mM Tris • HCl pH 7.4, 150 mM NaCl, 5% glycerol). Fractions containing pure MAS were collected and concentrated to 3 mg/mL using a 50k Amicon Ultra spin concentrator (Millipore) and flash frozen in aliquots for storage at −80 °C.

## MAS crosslinker loading and crosslinking

Crosslinkers 1, 2, or 3 (Supplementary Fig. 3) were loaded onto the ACP of apo-MAS by a one-pot chemoenzymatic method[25,60]. The optimized reaction included 94.7 μL buffer (50 mM Tris • HCl pH 7.4, 150 mM NaCl, 10% glycerol) 5 μL CoAA (1 mg/mL), 5 μL CoAD (1 mg/mL), 5 μL CoAE (1 mg/mL), 2.9 μL Sfp (6.9 mg/mL), 25 μL MgCl2 (250 mM), 2.5 μL DTT (100 mM),53.3 μL ATP (75 mM), and 0.88 μL 1, 2, or 3 (from a 10 mM DMSO stock). In this step, crosslinkers are converted into CoA analogs of 1, 2, or 3 (primed crosslinkers) by CoAA, CoAD, and CoAE as described in Supplementary Fig. 4B. The reaction was incubated at 23 °C for 2 h. Sequentially, 55.7 μL apo-MAS (31 μM) was added into the reaction mixture to generate a solution with 0.02 mg/mL CoAA, 0.02 mg/mL CoAD, 0.02 mg/mL CoAE, 0.08 mg/mL Sfp, 25 mM MgCl2, 1 mM DTT, 16 mM ATP, and 35 μM 1, 2, or 3, 7 μM MAS, and buffer (50 mM Tris • HCl pH 7.4, 150 mM NaCl, 10% glycerol). The reaction was then incubated at 37 °C for 1.5 h, followed by incubation at 23 °C for 18 h. The reaction mixture was then subjected to purification by HiLoad Superdex 200 pg (16 × 600) (GE Biosciences) size exclusion column with buffer (50 mM Tris • HCl pH 7.4, 150 mM NaCl). Fractions containing the crosslinked complexes were collected, diluted to 4.4 μM, and used immediately for cryo-EM sample preparation.

### Grid preparation for cryo-EM data collection

Holey carbon grids (Quantifoil, R1.2/1.3, Cu, 300 mesh) were glow discharged, and 3.2 μL of 1 mg/mL crosslinked MAS sample was applied to the grid in a Vitrobot Mk IV with the chamber at 4 °C and 95% humidity. The grids were blotted, plunge frozen in liquid ethane, and stored in liquid nitrogen until needed.

### Single particle cryo-EM data collection

Cryo-EM data were collected using a Titan Krios G4 operating at 200 kV and equipped with a Selectris X energy filter and a Falcon 4 direct electron detector. A total of 3135 movies were collected at a nominal magnification of 130,000×, nominal pixel size of 0.889 Å/px, and dose of 55 e⁻/Å² using the EPU software on a set of manually selected grid squares.

### General cryo-EM data processing

Movie data was imported into cryo-SPARC, and movie frame alignment was performed using patch motion correction, and patch CTF was used for CTF estimation. Exposures were curated to remove images with poor CTF fits, large in-frame motion, or extremely thick ice. This set of micrographs was used for all four maps reported in this study. All particle processing was carried out in cryo-SPARC, and final maps were sharpened in Phenix using *phenix.local_aniso_sharpen*. The initial stack of particles was picked using reference-free "blob" picking and subsequent 2D-classification to remove junk particles generated the initial stack of particles used in all subsequent operations. The resolution of all maps was estimated based on the gold standard Fourier shell correlation (FSC) using a cutoff of 0.143[61,62].

### Complex A cryo-EM data processing

Initial classes representing side views where the full MAS particle was visible were selected and used as templates to select a new stack of particles. These were subjected to 2D classification, where junk particles were removed. These particles were used to ab initio reconstruct 4 classes, and these classes were subsequently heterogeneously refined (~113k particles). The particles from the 3D class representing the full MAS dimer were 3D classified twice, first to remove remaining junk particles and second to select particles belonging to the doubly ACPs = KSs population (35,170 particles). This was non-uniformly refined, enforcing C2 symmetry with pose marginalization to generate a 3.87 Å reconstructed map for complex A[63].

### Complex B cryo-EM data processing

The full initial stack of cleaned particles (~184k particles) Fourier cropped to 1.78 Å/px were used in an ab initio 3D-reconstruction job with 3 classes, 1 class contained particles corresponding to the KS-LD-AT domains (complex B), 1 class contained particles corresponding to the DH-ΨKR-ER-KR domains (complex C), and 1 class captured additional junk particles. The class corresponding to the KS-LD-AT tridomain, when refined, showed strong map density for the first crosslink and some weaker density for a second crosslink on the other protomer. The particles were then extracted at full resolution and passed through 3D classification, and those corresponding to the double crosslink and those without any crosslinking were removed. The remaining particles (34,615) were non-uniformly refined in C1 symmetry, which generated a map for complex B at 3.22 Å.

### Complex C cryo-EM data processing

The ab initio class containing particles corresponding to the DH-ΨKR-ER-KR domains was reextracted at full resolution and passed through 3D classification to remove particles that did not have any crosslinking. The remaining particles (35,969) were non-uniformly refined in C1 symmetry, which generated a map for complex C at 3.74 Å. Density map demonstrated in all Complex C figures was generated by DeepEMhancer[64] with training models included in the package and

default tight masking through the half maps was generated from refinement as the input volumes.

### Complex D cryo-EM data processing

While processing the data for complex A, we observed a 3D class that corresponded to a single ACP = KS crosslink with a single ACP = DH crosslink. This map was used to generate templates, and particles were selected and extracted. The particles were passed through 2D classification to remove junk particles. We reasoned that the particle stacks for complexes B and C likely also contained some particles that corresponded to this crosslinking state, so those particle stacks were combined as well, duplicates were removed, and the full stack was used with the initial model from 3D classification in a homogeneous refinement job. Subsequent rounds of 3D classification were used to remove particles not corresponding to the ACP = KS/ACP = DH state. The final stack of particles (19,622) was non-uniformly refined in C1 symmetry to generate a 6.15 Å map for complex D.

### Structure modeling

The AlphaFold predicted structure of MAS from *M. bovis* (AF-Q02251) was used as the starting model for complexes A–C, the condensation and modifying compartments (except the ACPs) were fit individually into the maps using UCSF ChimeraX. The ACP domain model, also from AF-Q02251, was fit into the maps separately. The models were rejoined and manually remodeled in Coot. Density was observed for crosslinker **1** in all KS domains, including the shunt KS in complex B. Ligand density within the DH domain of complex C was limited. After modeling of **1**, the structures were refined using real-space refinement in Phenix. All map-to-model validation was also carried out in Phenix. AlphaFold structure of MAS from *M. bovis* (UniProt ID: Q02251) contains one amino acid difference compared with our biological sample, MAS from *M. tuberculosis* H37Rv (UniProt ID: I6Y231). Therefore, Cys213 from the AlphaFold model (UniProt ID: Q02251) was changed into Ser213 (UniProt ID: I6Y231) in the final model.

### Evaluating MAS mutation through screening crosslinking efficiency

We began by preparing a master mix of the coenzyme A analogue of **1** (Supplementary Fig. 4). This was conducted using 9.7 μL buffer (50 mM Tris • HCl pH 7.4, 150 mM NaCl, 10% glycerol), 0.5 μL CoaA (1 mg/mL), 0.5 μL CoaD (1 mg/mL), 0.5 μL CoaE (1 mg/mL), 0.07 μL Sfp (6.9 mg/mL), 2.5 μL MgCl₂ (250 mM), 0.5 μL DTT (100 mM), 2.7 μL ATP (75 mM), and 0.5 μL **1** (from a 1 mM DMSO stock). The reaction was incubated at 37 °C for 2 h. We then used this master mix for each reaction. Reactions were conducted by addition of 7.5 μL *apo*-MAS (13 μM) or 7.5 μL *apo*-MAS mutant (13 μM), stored at 23 °C for 2 h. These reactions generated mixtures containing 0.02 mg/mL CoaA, 0.02 mg/mL CoaD, 0.02 mg/mL CoaE, 0.02 mg/mL Sfp, 25 mM MgCl₂, 2 mM DTT, 8 mM ATP, and 20 μM **1**, 4 μM MAS or 4 μM MAS mutant, and buffer (50 mM Tris • HCl pH 7.4, 150 mM NaCl, 10% glycerol) at their final concentrations. The reaction was then incubated at 23 °C for an additional 2 h. The experiments were conducted in triplicate and evaluated by 6% SDS-PAGE. Using densitometry calculations by ImageJ[57], the crosslinking yield was determined by dividing crosslinked (>290 kDa) band intensity by the total of crosslinked (>290 kDa) and uncrosslinked (226 kDa) band intensity.

### Crosslinking excised ACP from MAS in trans

Crosslinking in trans between ACP and excised KS, excised KS-LD-AT, full MAS, or FabB follows a similar protocol. Crosslinkers were loaded onto excised *apo*-ACP by a one-pot chemoenzymatic method[25,60]. The reaction included 10.6 μL buffer (50 mM Tris • HCl pH 7.4, 150 mM NaCl, 10% glycerol), 29.4 μL excised ACP (0.17 mM), 0.5 μL CoaA (1 mg/mL), 0.5 μL CoaD (1 mg/mL), 0.5 μL CoaE (1 mg/mL), 1.4 μL Sfp (1.4 mg/mL), 2.5 μL MgCl₂ (250 mM), 0.5 μL DTT (100 mM), 2 μL ATP

(200 mM), and 2 µL **1, 2, 3, 4, 5, 6, 7**, or **8** (from a 10 mM DMSO stock) (Supplementary Fig. 5). The mixture contains the following components with final concentrations of 0.1 mM excised ACP, 1 mM DTT, 0.01 mg/mL CoaA, 0.01 mg/mL CoaD, 0.01 mg/mL CoaE, 0.04 mg/mL Sfp, 12.5 mM $MgCl_2$ 0.4 mM crosslinker, and 8 mM ATP. The reaction was incubated at 37 °C for 2 h. Excess probes were removed by 3 K MWCO Amicon Ultra spin concentrator (Millipore), resulting in a crude *crypto*-ACP with a concentration of 0.1 mM. Sequentially, 10 µL of this *crypto*-ACP (0.1 mM) mixture was added into 0.74 µL of excised KS (0.27 mM), excised KS-LD-AT (0.27 mM), full MAS (0.27 mM), or FabB (0.27 mM) in 9.2 µL buffer (50 mM Tris • HCl pH 7.4, 150 mM NaCl, 10% glycerol). The resulting reaction mixture containing 0.05 mM excised ACP, 0.01 mM KS containing protein (excised KS, excised KS-LD-AT, full MAS, or FabB) was incubated at 23 °C for 16 h. The in trans crosslinking was evaluated by 6% or 12% SDS-PAGE gel.

#### *E. coli* AcpP crosslinking with MAS and *E. coli* partner proteins

Crosslinking between *E. coli* AcpP and full MAS, or *E. coli* type II partner protein FabF (KS), FabF Cys163Ala (KS active site residue mutant), FabG (KR), FabA (DH), FabI (ER), FabD (AT), excised KS, and full MAS follows a similar protocol. The expression, purification, and preparation of *E. coli* type II *apo*-AcpP, *crypto*-AcpP, and partner proteins adopt the same protocol as reported[20]. Crosslinkers were loaded onto AcpP by a one-pot chemoenzymatic method[25]. This reaction included 0.1 mM AcpP, 1 mM DTT, 0.01 mg/mL CoaA, 0.01 mg/mL CoaD, 0.01 mg/mL CoaE, 0.04 mg/mL Sfp, 0.2 mM crosslinker **2, 3, 5, 6**, or **8**, and 8 mM ATP in buffer (50 mM Tris • HCl pH 7.4, 150 mM NaCl, 10% glycerol). These reactions were incubated at 23 °C for 21 h. The reaction mixture was then purified by a HiLoad Superdex 75 pg (16 × 600) (16 × 600) (GE Biosciences) size exclusion column through FPLC (ÄKTA Pure) using a buffer containing 50 mM Tris • HCl pH 7.4, 150 mM NaCl, 10% glycerol to yield pure *crypto*-AcpP loaded with crosslinker **2, 3, 5, 6**, or **8** (Supplementary Fig. 3). *Crypto*-AcpPs were concentrated to 0.1 mM by a 3 K MWCO Amicon Ultra spin concentrator (Millipore), flashed frozen, and stored at −80 °C. A crosslinking reaction contains the following components with their final concentrations of 0.05 mM *crypto*-AcpP with 0.01 mM FabF, 0.01 mM FabF Cys163Ala, 0.01 mM FabG, 0.01 mM FabA, 0.01 mM FabI, 0.01 mM FabD, 0.01 mM excised KS, or 0.01 mM full MAS. The crosslinking was evaluated by 6% or 12% SDS-PAGE gel (Supplementary Figs. 5–6).

#### Evaluation for the integrity of MAS and MAS mutants

Potential protein misfolding in MAS and MAS mutants were evaluated by labeling with fluorescent probe **13** (Fig. 3 and Supplementary Fig. 44). The labeling reaction contains a final concentration of 4 µM of MAS, 4 µM of MAS mutants, or 4 µM of denatured MAS with 0.02 mM **13**, 20 µM Sfp, 8 mM ATP, and 25 mM $MgCl_2$ in buffer (50 mM Tris • HCl pH 7.4, 150 mM NaCl, 5% glycerol). After 3 h of incubation at 25 °C, samples were evaluated with native 4–15% Mini-PROTEAN RGX (BioRad) gels at 150 V for 1 h. The resulting gels were imaged with a UV transilluminator at 365 nm (fluorescence) prior to Coomassie Brilliant Blue staining (total protein) (Supplementary Fig. 44).

#### SDS PAGE and native PAGE gel analyses

SDS PAGE analyses were conducted with in-house prepared 6% gels or commercially prepared gradient SDS PAGE. SDS PAGE gels (Supplementary Fig. 45) were developed to identify the crosslinked products. Full gel images are used for all figures unless otherwise noted, and copies of these gels are in Supplementary Fig. 46. Conventional running buffers were used unless noted otherwise.

#### Reporting summary

Further information on research design is available in the Nature Portfolio Reporting Summary linked to this article.

## Data availability

Single-particle cryo-EM maps for complexes A–D have been deposited in the Electron Microscopy Data Bank (EMDB) and the associated atomic model coordinates for complexes A-C in the Protein Data Bank (PDB) under accession codes EMDB-46504 and 9D2Y (complex A); EMDB-46505 and 9D2Z (complex B); EMDB-46506 and 9D30 (complex C); and EMDB-46507 (complex D). Summary data for these structures are presented in Supplementary Table 1. Source data are provided with this paper.

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

## Acknowledgements

The authors acknowledge the facilities of the UC San Diego cryo-EM facility, part of the Goeddel Family Technology Sandbox, along with the scientific and technical assistance of facility staff members Dr. Mariusz Matyszewski and Dr. Inga Kuschnerus. We also acknowledge J. Noel (Salk Institute), D. Lyumkis (Salk Institute), S. Konno (UC San Diego), D.O. Passos (Salk Institute), and A. Patel (UC San Diego) for initial studies on D.H. crosslinking[21] of MAS. This study was supported by NIH R01GM095970 to MDB and facilitated by a cryo-EM Seed Grant from the UC San Diego Department of Chemistry and Biochemistry. G.W.H. was supported by K12HL141956.

## Author contributions

Z.J., J.J.L., and M.D.B. designed the research. Z.J., G.W.H., and J.A.C. produced the samples. Z.J., J.A.C., and J.H. conducted crosslinking and mutational studies. Z.J. and G.W.H. prepared the samples for cryoEM. G.W.H. and Z.J. carried out the cryoEM data collection. Z.J., G.W.H., J.J.L., and M.D.B. analyzed the cryoEM data. J.J.L. and M.D.B. guided the project. M.D.B. supervised the project. Z.J., G.W.H., J.J.L., and M.D.B. wrote the manuscript with input from all authors.

## Competing interests

The authors declare no competing interests.
