## [Transparent Peer Review file · Nature Communications]

Visualizing Acyl Carrier Protein Interactions within a Crosslinked Type I Polyketide Synthase

Corresponding Author: Professor Michael Burkart

Version 0:

Reviewer comments:

Reviewer #1

(Remarks to the Author)

See attached file

Reviewer #2

(Remarks to the Author)

The manuscript by Jiang, Heberlig, et al reports cryo-EM structures of mycocerosic acid synthase from *Mycobacterium tuberculosis* using crosslinking probes to trap ACP into the catalytic domains KS and DH. Based on the multidomain complex structures of MAS and mutagenesis experiments, the authors propose a potential gating mechanism of KS for the iterative vectorial biosynthesis of MAS. The mechanism of domain-domain interactions in type I PKS is a critical bottleneck for understanding these enzyme machinery, and this study presents valuable structural analyses, particularly the symmetrical binding of both ACPs to the two KSs, revealing that the two reaction chambers formed by the PKS homodimer can work synchronously (and likely independently within the constrain of tilt and twist of the modifying compartment). Overall, the the manuscript presents solid structural data and enough supplementary figures to visually communicate the results. However, the gating mechanism of KS does not appear to be well-supported by the current provided discussions, and the manuscript writing contains many ambiguities that make it difficult to follow.

One of the core discussions of this work is the gating mechanism of KS. Burkart and co-workers have previously revealed the gating mechanism of KS in the type II bacterial FAS system (Ref. 31,32) based on the observations of both open- and closed-form of FabF/B-ACP complex structures. However, as shown in ExFig33, type II FAS and type I PKS do not share similar surface interaction mechanism and it should be cautious to use results obtained from the type II FAS analysis in supporting current result of MAS.

As reasoned in L229-233, the gating mechanism by the loop 1/2 is indirectly proposed based on the lack of turnstile vectorial mechanism and its positional similarity to the type II FAS KS-ACP interaction. However, the lack of "a turnstile vectorial mechanism (L230)" can also lead to other mechanisms, as seen in the Lsd14 structure (Ref.14), where Kim et al. proposed a pendulum mechanism. Turnstile mechanism is still not an established mechanism for modular PKSs (cf. Ref. 35), and current result does not provide any direct structural evidence to support the movement of this gating mechanism in MAS.

The authors provide site-directed mutagenesis result targeting residues on loop1/2, but the activity differences of crosslinking can only indicate their importance for domain-domain interaction, not the movement of them. Thus, while it is fair to discuss potential gating mechanism in MAS, I consider current results are not compelling enough to establish the gating mechanism as claimed in the abstract and main text.

Another issue critical for the vectorial iterative biosynthesis of MAS is the mechanism for substrate-specific elongation of the fully saturated chain. The loop1/2 of KS interact with the Ppant moiety of ACP-bound substrate, thus cannot distinguish the substrate difference. As mentioned in L274, preventing premature reloading (beta-keto, hydroxy, and olefin) is important but its mechanism remains unclear. Since it seems there are no architectural barrier like turnstile-closed state in type I PKS, the KS would need to prevent improper substrate elongation by specific substrate binding. An analysis of the substrate-binding

tunnel structure might provide information for such substrate specificity.

Also, structurally MAS resembles mammalian FAS and type I PKS more, and it would be informative to increase the discussion about their comparison. In DEBS PKS, it has been revealed that there are two different interaction modes of the ACP-KS interaction for chain translocation and the KS-ACP interaction for methylmalonyl-ACP condensation (<https://doi.org/10.1038/s41589-024-01709-y>). In iterative PKS like MAS, are there differences on the mode of ACP-KS interaction between translocation and condensation stages? How do the ACP-bound structures obtained here align with the non-cross-linked MAS structure in Ref.5?

The manuscript contains many misleading and ambiguous writing and the clarity can also be improved.

1. L19 “dual-site selective” and “dual site-selective” in L50
2. L32 enoyl reductase is more commonly described as enoylreductase in type I PKS.
3. L36 is “composed” of
4. L38 The enzymatic domains that catalyze chemical steps should be five.
5. L82 References for the published activities of KS are needed. Ref.5 only gives structural analysis but not activity.
6. L87 “no crosslinking between MAS and AcpP” ExFig5c shows weak bands at above 500 kDa.
7. L88 “PPI between ACP and KS are distinct in type I and II” but ExFig5A shows “mild” crosslinking between crypto-MAS-ACP and FabB.
8. L100 The transfer speed of PPant analogs from CoA to ACP should not change. The shunt product formation is minimized because all of free 1 were converted into CoA form first to avoid the contact of 1 and MAS.
9. L179 Not clear improved map density compared to what. Complex A also has ACP crosslinking so the improvement should not be due to the covalent ACP crosslinking as written in L180. Also recent work by Khosla to improve the resolution of structure and can be cited. <https://doi.org/10.1038/s41589-024-01709-y>
10. L185 “In protomer 2”.
11. L196 Interaction between ACP and “KS’ ” are limited to a single hydrogen bond—The KS’ and ACP shares significant surface for interaction and I do not think this interaction is only limited to a single hydrogen bond. Other interactions such as electrostatic interaction or hydrophobic interaction should also contribute to this domain domain interaction.
12. L207 “more interactions involving helix 1 and 2” Interactions indicating between ACP and KS’? Also, ACP-AT interaction is reported for DEBS ACP2(2) structure, which is a translocation ACP from upstream module.
13. L221 “undergo decarboxylative Claisen condensation”
14. L266 What does apo form of AT indicate?
15. L280 “pseudodimer” should be defined explicitly.
16. L295 “Thr2056” to be Tyr2056
17. L368-371 There are many “This” and what these this indicate is ambiguous.
18. L395 This sentence is ambiguous and needs careful reading to get the meaning. I would suggest revising or split into two or more sentences.
19. L404 “substrate specificity” should refer to how KS discriminate differently processed beta-keto intermediates, but I do not see different substrates (beta-keto, hydroxy, olefin, and methylene) are discussed in Ref.31.
20. L406 “negative cooperatively” should be defined explicitly.
21. L408 It is hard to understand the logic of how the methylmalonyl-CoA-selective AT affects the KS-substrate interaction
22. Fig1. there should be ACP-KS (translocation) between ACP-ER and ACP-AT. The upleft ACP-KS is at condensation stage, but the crosslinking structures should represent translocation stage as I understand.
23. Fig2b. the linkers’ length shown here do not match the main text such as L147, L162, etc. and the ACP with 300 amino acid in length looks too long.
24. ExFig4. Why was Sfp added in the first step, despite its function to be transferring CoA 1 into apo ACP? What is primed 1 exactly? Can crosslinker 1 directly react with ACP, so that R to be OH?
25. ExFig5. Could the authors explain how to interpret multiple crosslinking bands with different sizes?
26. ExFig7, 23 and others. What are b1-4?
27. ExFig22. The use of cerulenin and how to interpret these data are not discussed.
28. ExFig24 and others. What is the difference between panels A-C?
29. ExFig28. The crosslinking between KS and ACP substrate is not visible. The catalytic Cys in KS should be labeled.

Reviewer #3

(Remarks to the Author)

The authors present an impressive effort in determining the structure of the type I ketide synthase involving a particularly complex sample preparation and highly relevant biological impact. It is a well-presented paper with impressive documentation in the supplementary. The insights provided and experiments performed are of high quality and fit for publication in Nature communications, provided the authors address some concerns.

I have 2 major points and some minor points:

Major points

- The lack of discussion of oligomeric state and potentially hydrodynamic radius in solution in general hampers the analysis in several points (see also minor points). For example, when the authors suggest potential mechanisms and conformations that should be supported by experiments beyond a static look at a structure. Indeed the mutants that the authors characterize may or may not support this mechanism- “the correlation between crosslinking yield and enzymatic activity has been

validated". This is presented by the authors as a general law, but is it valid for these mutants that are underpinning the analysis of this specific structure and suggested mechanism? i.e. are these mutations impacting enzymatic activity.

- In the discussion, the authors should mention and discuss their structure also vis-à-vis the recently published structure of PKS1 and PKS2 from *E. chlorotica* by Schmidt and colleagues (<https://www.biorxiv.org/content/10.1101/2024.07.22.604177v1.full.pdf>) which presents a rocking mechanism for the catalytic cycle.

Minor points

Methods details

- A lot of the paper rests on previously established syntheses and site specificity properties. The methods section should include some details about how these were performed and established. E.g. "loading with a one-pot mechanism as previously established. Briefly, protein A" etc. etc
- The paper has no data availability section, and the authors did not provide PDB validation reports to the reviewer. Nevertheless, PDB validation reports could be recreated by uploading structures to the wwPDB server. Still, papers should have a data availability section with all accession codes relevant to the study and PDB validation reports should be provided.
- The conditions under which the reactions in extended data fig.4 are carried out are not detailed- concentrations, temperature, pH are all missing. I cannot find a methods section corresponding to this priming reaction. Some NMR or mass spec data showing that the priming does indeed generate the expected product would also be beneficial. The meaning of in situ in the phrase "we first converted the probes into the primed CoA analogs in situ" is also not clear.
- Size exclusion chromatography traces of the sample making up the first structure in the paper and in general size exclusion chromatography traces of final cryo-EM preparations should be shown, indicating which fractions were collected.

Biophysical characterization

- The paper is somewhat lacking in biophysical characterization, hampering the interpretation of the model. For example:
- Pag. 5 "No crosslinking was observed between the crypto-ACP and KS or KS-LD-AT (Extended Data Figs. 5a-b). This is likely due to inactivation of the KS, as it has been reported that the excised MAS-like PKS KS-LD-AT tridomain construct exists as a monomer-dimer equilibrium in solution." The authors purified these constructs with size exclusion chromatography, and should be able to prove or disprove this statement. The authors should include size exclusion traces with Mw estimates.
 - "The full MAS structure adopts a homodimeric architecture (Fig. 2a), comprising two catalytic chambers (chambers 1 and 2, Fig. 2a)". Clearly, it is a homodimer in the structure that was solved (complex A), but what was the oligomeric state distribution in solution? SEC or mass photometry data should be shown.
 - The crosslinks observed in cryo-EM are likely a small subset of the ones actually formed in solution, i.e. they only comprise those that can be refined. Have the authors considered performing mass spectrometry to characterize the crosslinking, or at least the modification of the various proteins? I would challenge the statement of site-specificity presented in this case, as the authors do not present sufficient evidence that in the case of MAS these crosslinkers are site-specific. I apologise if the evidence is presented in previous work.

Model building

- The map provided for complex A does not look like a 3.87 angstrom resolution map: the details expected at this resolution are not present in the map. Is the map provided unsharpened? The authors should provide the map-to-model FSc and maybe also a locally filtered map for the higher resolution details in the paper.
- Looking at the models of complex A and B, secondary structures are not assigned and running dssp shows some strange deformations. This may be due to the authors doing real space refinement without applying secondary structure restraints. Re-refining in REFMAC or Phenix may improve results.

Text and data availability

- Some extended data figures are labelled as "extended data figure 3" and others as "extended data figure S5"
- Cloning section should include UNIPROT IDs of the proteins involved.
- Uncropped gel images should be source data, not supplementary figures.
- "Since Size exclusion chromatography afforded crosslinked MAS products at >45% crosslinking yield by SDS-PAGE (right, Extended Data Fig. 7)." Meaning not clear.
- I am not sure that the section justifying the choice of C16- α -bromoamide 1 based on in trans crosslinking of *E. coli* ACP makes sense: why would this system be an appropriate model for *M. tuberculosis* MAS? I understand the point of conformational flexibility, but isn't *E. coli* ACP chosen simply because of its in trans properties?
- The whole paragraph starting with "Evaluating the resolved crosslinked structures across.." feels more like a discussion section and should perhaps be shortened. At this point, the fact that 3d classification in cryo-EM can provide multiple conformations is not really something that needs mentioning as a novel finding.
- Ψ KR abbreviation is introduced before its explanation. In general, reducing the use of domain acronyms would greatly improve readability. Please label domains in figure 1, instead of introducing domain naming/color scheme in figure only to then have the reader look backwards to understand the cycle in Fig.1 .
- "The LD has been previously reported to be involved in a turnstile mechanism, where a closed conformation forms extensive interactions with AT to block access to the KS." Citation missing

Version 1:

Reviewer comments:

Reviewer #1

(Remarks to the Author)
See attached files

Reviewer #2

(Remarks to the Author)

This revised manuscript has addressed most of my concerns and been improved in the discussion and interpretation of the structural data. The structural data presented here is indeed informative and deepens our understanding of the PKS enzymology. I only have a few follow-up comments.

1. L217-222 The comparison of MAS with modular type I PKSs are provided here, and it would be helpful to explicitly state that the DEBS-M1 and Lsd14 structures are at condensation state (KS-ACP within the same module) . In contrast, the DEBS-M3 and ACP2 structure is at the substrate reloading state (ACP_n and. KS_{n+1}), which is the identical state to the MAS observed structure.

2. L389 "allowing the ACP to interact with KS while crosslinking with KS" Does this describe the two ACPs? The ACP should not interact with KS from the same protomer.

3. L436 While I appreciate these discussions about substrate recognition, the TNGQ or VMYH motifs are only useful for modular cis-AT PKSs; for example in trans-AT PKSs, they do not have this conserved motifs (<https://doi.org/10.1016/j.str.2013.12.016>). Would it be possible to discuss it based on structure? For instance Fig. 3 shows substrate binding at KS, but it is unclear how the substrate binding pocket (or tunnel) recognizes (or interact with) the alpha,beta-position of the substrate.

Reviewer #3

(Remarks to the Author)

The authors have addressed my concerns and I recommend the paper for publication. However, the supplementary figures of the size exclusion chromatography runs have weird artefacts from copy pasting unloaded images, or something like that. They are not displayed properly.

Version 2:

Reviewer comments:

Reviewer #1

(Remarks to the Author)
See attached file

Reviewer #2

(Remarks to the Author)

The authors have addressed all my concerns and I appreciate their efforts for this revision.

The manuscript describes structures of *M. tuberculosis* mycocerosic acid synthase (MAS), a type I iterative polyketide synthase (PKS) that produces C6-C20 fatty acids. The 256-kDa polypeptide includes the full constellation of 5 enzyme domains for extension (KS, AT) and modification (KR, DH, ER) of an intermediate as well as an acyl-carrier protein (ACP) domain. The overall organization of dimeric MAS is similar to the metazoan type I fatty acid synthase (FAS) and also the type I iterative, fully reducing PKS proteins. The ACP phosphopantetheine (Ppant) cofactor was armed with a series of mechanism-based crosslinkers that trapped the ACP at either the KS catalytic Cys or the DH catalytic His.

Atomic models built into cryo-EM maps are provided for 3 structures (A-C)

(A) full MAS dimer with ACPs crosslinked to KS active sites

(B) dimer of KS-AT extension region with 1 ACP crosslinked to 1 KS

(C) dimer of DH-ER-KR modifying region with 1 ACP crosslinked to 1 DH

The resolution of a fourth map was too low for model building

(D) low-res map of a full MAS dimer with 1 ACP crosslinked to KS & 1 ACP crosslinked to DH

Trapping the ACPs provided greater detail of ACP-KS and ACP-DH interactions, and higher resolution than for the uncrosslinked enzyme or were previously available. The structures reveal several MAS conformations with substantial differences in the inter-domain hinges, thus demonstrating a remarkable PKS flexibility.

The results have relevance to other systems with similar architecture. In particular, the conformational variability associated with ACP localization in MAS may be a model for analogous conformational contortions of metazoan FAS, where the catalytic domains are more flexibly linked. The results are also relevant to other type I iterative PKS (for example the well studied LovB of lovastatin biosynthesis) and, to a lesser extent, the modular type I PKS proteins (where the domain contacts differ in the modifying region, and the ER has no dimer contacts).

Most of the key regions of the EM maps support the models provided by the authors. This is critical for the linkers between KR and ACP domains, which are necessarily highly flexible, and for the crosslinks from ACP-Ppant to catalytic Cys or His. The cryo-EM maps provide strong support for these aspects of the structures. However, the density is ambiguous for the linkages of the KS-AT extension region to the DH in the modifying region. The extension-modification connection is modeled only for Complex A, which includes the entire MAS. The “post-AT linker” exits the extension region near the molecular twofold. The polypeptide transits directly to the twofold where it meets the partner protomer in a snarl of density before entering the DH. The path of the polypeptide through this density snarl is not resolvable at any contour level, so it is impossible to know which extension region (KS-AT) and which modifying region (DH-ER-KR) belong to a single protomer. In fact, in the Complex A model the polypeptides are intertwined: they cannot be separated without dissociating the dimer (see images below). The snarl of density at the molecular twofold between the DH domains also exists in the Complex C map (where only the DH-ER-KR modifying region is modeled) and the Complex D map (no model).

The SDS PAGE gels show that inter-subunit crosslinking can occur, but do not demonstrate that this is the exclusive interaction mode of ACP and KS. Thus the claim of ACP(2) crosslinked to KS(1), and vice versa, is not supported by the presented data.

It would be helpful to list inter-domain hinge differences among the 4 Complexes A-D

Nowhere in the text or SI is the biological source of MAS identified. An AlphaFold model of the *M. tuberculosis* var. bovis MAS sequence was used for initial atomic models. However, Table S2 in the cited paper from Hugo Gramajo, who provided the clone, lists MAS from strain *M. tuberculosis* H37Rv. *M. tuberculosis* strain and accession number for the protein sequence should be included.

Technical comments (streamed through my initial read of the manuscript):

Fig. S1: FadD28 reaction is 5'-adenylation, not phosphorylation

Fig. S1: indicate in caption that gray-ed areas in Complexes B & C are present, but disordered

Fig. S7: label the lanes; define SDS gel bands b1, b2, b3, b4 here (not Fig. S45)

Table S1: B factors for ligands (0.5 \AA^2) are ridiculous – typo or Phenix bug??

Figs. S8,9: include particle numbers with map images

Fig. S13: show density for inter-domain linkers – specify density contour level

Fig. S14: panel B is Complex A? (caption says B); add density contour level in caption

Protomers 1 & 2 vs. protomers A & B – confusing – stick with primed KS' & unprimed KS

Compare overall conformations of Complexes A & B

Figs. S10-13,15,16: Label the primed & unprimed domains

Fig. S18: show small-molecule crosslinker density for 2nd subunit of Complex B

Text lines 278-286: DH has 2 catalytic residues (His929 & Asp1095) so both halves of the double hotdog contribute to catalysis (His & Asp are on opposite monomers of type II FabA). Lose the “pseudomonomer” nomenclature.

Fig. S22: would help readers, especially the color-blind, if the labels were below the data bars at a 45° angle.

Fig. S34C: “pseudoheterodimer” is unclear – type II FabA/FabZ have a dimer of the hotdog fold; type I PKS/FAS have a dimer of the double-hotdog fold; use this typical nomenclature. Would be helpful to include Asp1095 in this figure.

Fig. S36: specify density contour level in caption

Fig. 4b caption: Asn – Thr interaction may be a hydrogen bond, it's not a salt bridge

Fig. 4d,e,f: cannot see the point (no density, domain movement relative to what?, tunnel invisible) – caption should refer readers to the SI figures with greater detail of Fig 4 panels

Fig. S37B: E. coli FabA is specific for C10 substrates (FabZ acts on the named substrates)

Fig. S39: what was superimposed? (panel A appears to be entire polypeptide) – panels B-D focus is on DH-DH hinging, so should be superposition of 1 DH subunit – (include RMSDs for all). RMSD values should be in the caption.

Fig. S41 both panels: label the residues where measurements were taken

Fig. 5: which is chamber 1, chamber 2? Which side of dimer is KS and which KS'? – label

The authors have responded in substance to all comments from this reviewer. Many points are clearer and better illustrated in the revised manuscript, figures and SI.

One major point remains. The EM map density is ambiguous for the dimer neck that connects the elongation region (KS-LD-AT) to the modifying region (DH-ΨKR-ER-KR-ACP). Thus, how these regions are connected within a protomer cannot be assigned in the structures. Moreover, while the crosslinking data demonstrate that crosslinks can form between protomers (ACP to KS), the data are also consistent with intra-protomer crosslinking. A major band labeled “uncrosslinked” or “b1” is prominent in all denaturing gels of crosslinked protein and likely includes proteins with intra-protomer crosslinks (Fig. 2C & Figs. S7, S23-S27, S45). The mixture of species existed in the sample used for EM.

The authors responded to my comment about the structure and crosslinking ambiguity by removing 12 amino acids from the linker1 region of the models, and by inserting text indicating that they cannot exclude intra-protomer crosslinking. This is a good response, but it does not go far enough because the interpretation of inter-protomer crosslinking is baked into the text and figures.

I suggest a simple relabeling of domains in the text and the figures. The protomers and domains of the MAS dimer are labeled as unprimed or primed (KS, KS', ACP, ACP', etc.). However, it is unknown whether the domains labeled KS-LD-AT are in the same protomer as those labeled DH-ΨKR-ER-KR-ACP or in the protomer labeled DH'-ΨKR'-ER'-KR'-ACP'. If the labels KS-LD-AT and KS'-LD'-AT' are retained for the elongation region of the MAS protomers, then the domains of the modifying region could be labeled $DH^A-\Psi KR^A-ER^A-KR^A-ACP^A$ and $DH^B-\Psi KR^B-ER^B-KR^B-ACP^B$. Points in the manuscript file where the inter-protomer interpretation should be changed are highlighted in the file I return with this review. This additionally applies to several of the supplemental figures and captions (10-16, 18-21, 28-29, 40-41, 43, 48, 51-52).

None of this affects the important and well supported conclusions of the study. Together the crosslinking and structural data show that the ACP can deliver substrates to the KS active site in the opposite protomer of the MAS dimer, but NOT that this is the exclusive mode of delivery. This is important because many structure-based models have been proposed for PKS throughput, most with cross-protomer catalysis, but no one has yet shown that this is the predominant means of catalysis. In contrast to ACP-KS interaction, the data here indicate that any ACP-DH crosslink is predominantly/exclusively intra-protomer: no inter-protomer crosslink when the KS catalytic Cys was substituted with Ala (Fig. S26), but the DH catalytic His was available.

The study reveals remarkable inter-domain flexibility in PKS systems. The crosslinking trapped the ACP in relevant positions that also stabilized multiple conformations of the MAS dimer. This is an important study of broad interest. But please do not perpetuate models for catalytic throughput that have not (yet?) been demonstrated.

The manuscript text and figure labeling now address my comments about the connectivity of the condensing (KS-LD-AT) and modifying (DH-ER-KR) regions of the MAS megasynthase. The manuscript presents significant results that will be of great interest and import to PKS researchers. Only one small change is needed. In the final figure showing the model for catalytic throughput, the condensing-modifying linkers are shown between particular domains of MAS (left image below). In fact, these connectivities cannot be established in the EM maps of this study (or any preceding study). On the right, I crudely drew connectivities that reflect what is visible in the authors' EM maps, i.e. the linkers join in a knot of density just below the green DH domains. I suggest that in all MAS images in Figure 5, either the gray linker lines be eliminated altogether, or – preferentially – be drawn to reflect the ambiguity in their maps.

The very nice Figure 5 summarizes the important finding that MAS undergoes substantial tilting and twisting of the upper modifying region relative to the lower condensing region. Please do not perpetuate as-yet-unproven ideas about connectivity!

Nature Communications
NCOMMS-24-72772-T
Response to Reviewer Comments

Item 1, Reviewer 1: The “post-AT linker” exits the extension region near the molecular twofold. The polypeptide transits directly to the twofold where it meets the partner protomer in a snarl of density before entering the DH. The path of the polypeptide through this density snarl is not resolvable at any contour level, so it is impossible to know which extension region (KS-AT) and which modifying region (DH-ER-KR) belong to a single protomer. In fact, in the Complex A model the polypeptides are intertwined: they cannot be separated without dissociating the dimer (see images below). The snarl of density at the molecular twofold between the DH domains also exists in the Complex C map (where only the DH-ER-KR modifying region is modeled) and the Complex D map (no model).

Response: We agree that crosslinking within the same protomer should not be excluded. In response, we deleted modeled residues 882-893 (part of the “post-AT linker” or “linker 1”) and mention this in lines 162-163. We have also added explanation in lines 126-128, 145-146, 186-187, and 383-386, emphasizing that this data does not exclude intra-protomer crosslinking.

Item 2, Reviewer 1: The SDS PAGE gels show that inter-subunit crosslinking can occur, but do not demonstrate that this is the exclusive interaction mode of ACP and KS. Thus, the claim of ACP(2) crosslinked to KS(1), and vice versa, is not supported by the presented data.

Response: See Item 1. We agree that crosslinking within the same protomer should not be excluded and have deleted modeled residues 882-893 (part of the “post-AT linker” or “linker 1”).

Item 3, Reviewer 1: It would be helpful to list inter-domain hinge differences among the 4 Complexes A-D.

Response: To address this comment, extended Data Fig. 52 has been added. Lines 360-365 are also added to discuss about the comparison of inter-domain linkers among complexes A-C. Complex D was not discussed because we did not perform atomic modeling for it due to lower resolution.

Item 4, Reviewer 1: Nowhere in the text or SI is the biological source of MAS identified. An AlphaFold model of the *M. tuberculosis* var. *bovis* MAS sequence was used for initial atomic models. However, Table S2 in the cited paper from Hugo Gramajo, who provided the clone, lists MAS from strain *M. tuberculosis* H37Rv. *M. tuberculosis* strain and accession number for the protein sequence should be included.

Response: This problem has been addressed in the **Structure modeling** section in the SI, with the following sentences added. “AlphaFold structure of MAS from *M. bovis* (UniProt ID: Q02251) contains one amino acid difference compared with our biological sample MAS from *M. tuberculosis* H37Rv (UniProt ID: I6Y231). Therefore, Cys213 and Cys213’ from the AlphaFold model (UniProt ID: Q02251) were changed into Ser213 and Ser213’ (UniProt ID: I6Y231) in the final model.”

Item 5, Reviewer 1: Fig. S1: FadD28 reaction is 5’-adenylation, not phosphorylation

Response: Revised as suggested. The reaction for FadD28 in Extended Data Fig. 1 is now 5’-adenylation.

Item 6, Reviewer 1: Fig. S1: indicate in caption that gray-ed areas in Complexes B & C are present, but disordered

Response: To address this comment, the following sentence was added to the caption of Extended Data Fig. 2, “Domains with grey shading are present but disordered.”

Item 7, Reviewer 1: Fig. S7: label the lanes; define SDS gel bands b1, b2, b3, b4 here (not Fig. S45)

Response: Revised as requested. The caption of Extended Data Fig. 7 is changed to “SDS-PAGE (6%) demonstrates the MAS crosslinking in cis using crosslinkers 1 and 2. Lanes 2-4 and 5-6 are consecutive fractions from FPLC purification. Lane 9 was selected for single particle analysis. The purified crosslinked complexes were identified as four bands, b1, b2, b3, and b4. Complex C and uncrosslinked MAS monomer were assigned to b1. Complex B was assigned to b2, and complexes A and D were assigned to b3. Intermolecular crosslinking of MAS (oligomer after denaturing) was observed as b4. A throughout analysis of b1, b2, b3, and b4 can be found in Extended Data Fig. 45.”

Item 8, Reviewer 1: Table S1: B factors for ligands (0.5 Å²) are ridiculous – typo or Phenix bug??

Response: Thank you for catching this Phenix bug, where the B-factors defaulted to 0.5. The B-factors have been recalculated and the updated in the manuscript.

Item 9, Reviewer 1: Figs. S8,9: include particle numbers with map images

Response: Revised as requested.

Item 10, Reviewer 1: Fig. S13: show density for inter-domain linkers – specify density contour level

Response: Revised as suggested. Extended Data Fig. 52 has been added. Brief guidance regarding this figure has been added to lines 161, 165, and 171.

Item 11, Reviewer 1: Fig. S14: panel B is Complex A? (caption says B); add density contour level in caption

Response: This typo has been addressed, and a contour level has been added to the caption.

Item 12, Reviewer 1: Protomers 1 & 2 vs. protomers A & B – confusing – stick with primed KS' & unprimed KS

Response: Apologies for the confusion. Protomers A and B have been replaced by protomers 1 and 2.

Item 13, Reviewer 1: Compare overall conformations of Complexes A & B

Response: To address this comment, Extended Data Fig. 51 was added to the SI and a guide to this figure has been added to line 422 in the Discussion section of the manuscript.

Item 14, Reviewer 1: Figs. S10-13,15,16: Label the primed & unprimed domains

Response: Revised as suggested. Domains in Extended Data Fig. 10-13, 15, 16, 31, and 32 are now labeled.

Item 15, Reviewer 1: Fig. S18: show small molecule crosslinker density for 2nd subunit of Complex B

Response: Revised as suggested. Extended Data Fig. 19 has demonstrated the small molecule crosslinker density for the second subunit of Complex B.

Item 16, Reviewer 1: Text lines 278-286: DH has 2 catalytic residues (His929 & Asp1095) so both halves of the double hotdog contribute to catalysis (His & Asp are on opposite monomers of type II FabA). Lose the “pseudomonomer“ nomenclature.

Response: Revised as suggested. Here, “pseudomonomer” has been changed to “double-hotdog fold” throughout the manuscript and SI. Only one catalytic pair of His797 and Asp1095 are present on one side of the double-hotdog fold, as indicated in Extended Data Fig. 34.

Item 17, Reviewer 1: Fig. S22: would help readers, especially the color-blind, if the labels were below the data bars at a 45° angle.

Response: Revised as suggested. The labels are now underneath the charts.

Item 18, Reviewer 1: Fig. S34C: “pseudoheterodimer” is unclear – type II FabA/FabZ have a dimer of the hotdog fold; type I PKS/FAS have a dimer of the double-hotdog fold; use this typical nomenclature. Would be helpful to include Asp1095 in this figure.

Response: See Item 16. All analysis of the “pseudoheterodimer” has been changed to “double-hotdog fold”. Asp1095 and more labeling have been added to Extended Data Fig. 34.

Item 19, Reviewer 1: Fig. S36: specify density contour level in caption

Response: Revised as suggested. Contour level is now indicated in the caption.

Item 20, Reviewer 1: Fig. 4b caption: Asn – Thr interaction may be a hydrogen bond, it’s not a salt bridge

Response: Revised as suggested. In Fig. 4b caption, “salt bridge” has been replaced by “hydrogen bond”.

Item 21, Reviewer 1: Fig. 4d,e,f: cannot see the point (no density, domain movement relative to what?, tunnel invisible) – caption should refer readers to the SI figures with greater detail of Fig 4 panels

Response: Revised as suggested. Fig 4d has been edited for clarity, and guidance to the Extended Data Figures has been added in the Fig. 4 caption.

Item 22, Reviewer 1: Fig. S37B: E. coli FabA is specific for C10 substrates (FabZ acts on the named substrates)

Response: Revised as suggested. The named substrates have been changed to “β-capric acid”.

Item 23, Reviewer 1: Fig. S39: what was superimposed? (panel A appears to be entire polypeptide) – panels B-D focus is on DH-DH hinging, so should be superposition of 1 DH subunit – (include RMSDs for all). RMSD values should be in the caption.

Response: Revised as suggested. MAS-like PKS DH’ and complex A DH’ were aligned to DH’ of complex C with RMSD= 1.568. Related discussion regarding this change has been detailed in lines 330-339 of the main text, Fig. 4 caption, and Extended Data Fig. 39.

Item 24, Reviewer 1: Fig. S41 both panels: label the residues where measurements were taken

Response: Revised as suggested. Residues have been labeled in Extended Data Fig. 41 and the measurements were detailed in the caption.

Item 25, Reviewer 1: Fig. 5: which is chamber 1, chamber 2? Which side of dimer is KS and which KS? – label

Response: Revised as suggested. Labeling has been added to Fig. 5.

Item 26, Reviewer 2: One of the core discussions of this work is the gating mechanism of KS. Burkart and co-workers have previously revealed the gating mechanism of KS in the type II bacterial FAS system (Ref. 31,32) based on the observations of both open- and closed-form of FabF/B-ACP complex structures. However, as shown in ExFig33, type II FAS and type I PKS do not share similar surface interaction mechanism, and it should be cautious to use results obtained from the type II FAS analysis in supporting current result of MAS.

Response: We agree that care should be taken in making these comparisons. Statements about KS gating mechanism have been rewritten or removed in lines 23, 226, 250, 257-260, 287, 390, 429, and 478.

Item 27, Reviewer 2: As reasoned in L229-233, the gating mechanism by the loop 1/2 is indirectly proposed based on the lack of turnstile vectorial mechanism and its positional similarity to the type II FAS KS-ACP interaction. However, the lack of “a turnstile vectorial mechanism (L230)” can also lead to other mechanisms, as seen in the Lsd14 structure (Ref.14), where Kim et al. proposed a pendulum mechanism. Turnstile mechanism is still not an established mechanism for modular PKSs (cf. Ref. 35), and current result does not provide any direct structural evidence to support the movement of this gating mechanism in MAS.

Response: See Item 26. Statements about KS gating mechanism have been rewritten or removed in lines 23, 226, 250, 257-260, 287, 390, 429, and 471.

Item 28, Reviewer 2: The authors provide site-directed mutagenesis result targeting residues on loop1/2, but the activity differences of crosslinking can only indicate their importance for domain-domain interaction, not the movement of them. Thus, while it is fair to discuss potential gating mechanism in MAS, I consider current results are not compelling enough to establish the gating mechanism as claimed in the abstract and main text.

Response: See Item 26. We agree that care must be taken in these comparisons. Therefore, statements about KS gating mechanism have been rewritten or removed in lines 23, 226, 250, 257-260, 287, 390, 429, and 478.

Item 29, Reviewer 2: Another issue critical for the vectorial iterative biosynthesis of MAS is the mechanism for substrate-specific elongation of the fully saturated chain. The loop1/2 of KS interact with the Ppant moiety of ACP-bound substrate, thus cannot distinguish the substrate difference. As mentioned in L274, preventing premature reloading (beta-keto, hydroxy, and olefin) is important but its mechanism remains unclear. Since it seems there are no architectural barrier like turnstile-closed state in type I PKS, the KS would need to prevent improper substrate elongation by specific substrate binding. An analysis of the substrate-binding tunnel structure might provide information for such substrate specificity.

Response: Revised as suggested. More detailed analysis of the MAS-KS substrate gatekeeping role is added in the discussion section lines 427-442.

Item 30, Reviewer 2: Also, structurally MAS resembles mammalian FAS and type I PKS more, and it would be informative to increase the discussion about their comparison. In DEBS PKS, it has been revealed that there are two different interaction modes of the ACP-KS interaction for

chain translocation and the KS-ACP interaction for methylmalonyl-ACP condensation (<https://doi.org/10.1038/s41589-024-01709-y>). In iterative PKS like MAS, is there differences on the mode of ACP-KS interaction between translocation and condensation stages? How do the ACP-bound structures obtained here align with the non-cross-linked MAS structure in Ref.5?

Response, Reviewer 2: These are important considerations. In response, a sentence in lines 171-173 has been added in the manuscript to navigate the readers to the new Extended Data Figs. 48 and 49. The complexes achieved in this study were achieved by using crosslinker **1**, which mimics the translocation step (substrate loading or reloading on KS). Therefore, we did not observe the expected condensation step conformation. In terms of the comparison between MAS complex B (in this study) and the condensation compartment of MAS-like PKS (Ref.5), the active site of MAS-like PKS are disordered as it was crystallized in a monomeric form (see lines 271-275). For the comparison between MAS complex C and the modification compartment of MAS-like PKS, there is a significant DH domain rotation upon crosslinking in MAS complex C which might play a role in shielding the active site of the DH for the vectorial biosynthesis (see lines 328-339).

Item 31, Reviewer 2: L19 “dual-site selective” and “dual site-selective” in L50

Response: Apologies for the typographical error. “Dual-site selective” has been changed into “dual site-selective” throughout the manuscript.

Item 32, Reviewer 2: L32 enoyl reductase is more commonly described as enoylreductase in type I PKS

Response: Revised as suggested. “enoyl reductase” has been changed to “enoylreductase”.

Item 33, Reviewer 2: L36 is “composed” of

Response: Revised as suggested.

Item 34, Reviewer 2: L38 The enzymatic domains that catalyze chemical steps should be five.

Response: Revised as suggested.

Item 35, Reviewer 2: L82 References for the published activities of KS are needed. Ref.5 only gives structural analysis but not activity.

Response: Revised as suggested. “This interpretation is supported by published activities of KSs in both type I and type II systems...” has been changed to “This interpretation is supported by published observation of KSs in both type I and type II systems...”

Item 36, Reviewer 2: L87 “no crosslinking between MAS and AcpP” ExFig5c shows weak bands at above 500 kDa.

Response: Revised as suggested. “showed no crosslinking with the same panel...” has been changed to “showed only minimal crosslinking with the same panel”

Item 37, Reviewer 2: L88 “PPI between ACP and KS are distinct in type I and II” but ExFig5A shows “mild” crosslinking between *crypto*-MAS-ACP and FabB.

Response: As shown in Extended Data Fig. 17, the interface between ACP and KS is distinct in type I and type II systems in terms of electrostatics. If MAS-ACP interacts with FabB efficiently, a high crosslinking yield can be expected, as shown by the crosslinking between *crypto*-AcpP and FabF in Extended Data Fig. 6. It is also worth noting that α -bromamide crosslinkers, such as crosslinker **1**, are highly reactive. With proper PPI, the completion of crosslinking can be achieved

within seconds, as indicated in Ref. 33. Therefore, the “mild” crosslinking between *crypto*-MAS-ACP and FabB demonstrates the highly reactive warhead of crosslinker 1 and a mild PPI compatibility between MAS-ACP and FabB.

Item 38, Reviewer 2: L100 The transfer speed of PPant analogs from CoA to ACP should not change. The shunt product formation is minimized because all of free 1 were converted into CoA form first to avoid the contact of 1 and MAS.

Response: Revised as suggested. “In this way, this optimized reaction sequence minimizes interaction time between the free probe and the *apo*-MAS, limiting shunt product formation.” has been changed to “In this way, this optimized reaction sequence minimizes interaction time between the free probe and the *apo*-MAS, limiting shunt product formation.”

Item 39, Reviewer 2: L179 Not clear improved map density compared to what. Complex A also has ACP crosslinking so the improvement should not be due to the covalent ACP crosslinking as written in L180. Also, recent work by Khosla to improve the resolution of structure and can be cited. <https://doi.org/10.1038/s41589-024-01709-y>

Response: Revised as suggested. The statement has been rewritten from line 187-191. This citation has been added.

Item 40, Reviewer 2: L185 “In protomer 2”.

Response: See Item 12. “protomer A” and “protomer B” have been changed into “protomer 1” or “protomer 2”.

Item 41, Reviewer 2: L196 Interaction between ACP and “KS’ ” are limited to a single hydrogen bond—The KS’ and ACP shares significant surface for interaction, and I do not think this interaction is only limited to a single hydrogen bond. Other interactions such as electrostatic interaction or hydrophobic interaction should also contribute to this domain interaction.

Response: Good point. Discussion regarding to the hydrophobic and electrostatic interactions were added to lines 206-208, with guidance to the newly added Extended Data Fig. 50.

Item 42, Reviewer 2: L207 “more interactions involving helix 1 and 2” Interactions indicating between ACP and KS’? Also, ACP-AT interaction is reported for DEBS ACP2(2) structure, which is a translocation ACP from upstream module.

Response: Revised as suggested. The addition of, “However, the observed ACP-AT interactions are in agreement with the recent complex between DEBS-M3 and ACP2 structure.” is now added, with citation in lines 216-221.

Item 43, Reviewer 2: L221 “undergo decarboxylative Claisen condensation”

Response: Revised as suggested. “undergo Claisen-like chain elongation” has been changed to “undergo decarboxylative Claisen condensation”.

Item 44, Reviewer 2: L266 What does apo form of AT indicate?

Response: Revised as suggested. A brief description “(without substrate in the active site)” has been added to the sentence in lines 278-279.

Item 45, Reviewer 2: L280 “pseudodimer” should be defined explicitly.

Response: See item 16. “Pseudodimer” related discussion has been replaced by “double-hotdog fold”.

Item 46, Reviewer 2: L295 “Thr2056” to be Tyr2056

Response: Apologies for the typo. “Thr2056” has been changed to “Tyr2056”.

Item 47, Reviewer 2: L368-371 There are many “This” and what these this indicate is ambiguous.

Response: Revised as suggested. The corresponding sentences have been changed to “Importantly, complexes A and B are the first validation of cross-protomer processivity in type I PKSs. Complex D shows that while ACP interacts with DH in chamber 1, ACP’ interacts with the KS in chamber 2 (Fig. 5) corroborating the obligate dimerization of MAS. The cross-protomer processivity plays an important role guiding synthetic efficiency in type I PKSs, and likely FASs by allowing the ACP to interact with KS while crosslinking with KS’.” Lines 383-389.

Item 48, Reviewer 2: L395 This sentence is ambiguous and needs careful reading to get the meaning. I would suggest revising or split into two or more sentences.

Response: Revised as suggested. The corresponding sentence has been changed to “In modular PKS Lsd14, the post-ACP dimerization element or pre-KR dimerization element keeps both ACP and ACP’ in proximity. 14,42,43 Due to this, the catalytic chambers function asynchronously in their proposed pendulum model. The absence of post-ACP dimerization element or pre-KR dimerization element allows MAS to function synchronously and asymmetrically.” Lines 418-422.

Item 49, Reviewer 2: L404 “substrate specificity” should refer to how KS discriminate differently processed beta-keto intermediates, but I do not see different substrates (beta-keto, hydroxy, olefin, and methylene) are discussed in Ref.31.

Response: Agreed. This has been addressed in lines 427-442.

Item 50, Reviewer 2: L406 “negative cooperatively” should be defined explicitly.

Response: Revised as suggested. This has been addressed in the 4th paragraph of the discussion session as well as Extended Data Fig. 43. Lines 444-449.

Item 51, Reviewer 2: L408 It is hard to understand the logic of how the methylmalonyl-CoA-selective AT affects the KS-substrate interaction

Response: Revised as suggested. “After the condensation step at the KS domain, the elongated substrate is barred from reversing, as the AT, although proximal, strictly accepts methylmalonyl-CoA.” has been changed to “After the condensation step at the KS domain, the elongated substrate is barred from revisiting AT, as the AT, although proximal, strictly accepts methylmalonyl-CoA.” Lines 449-451.

Item 52, Reviewer 2: Fig1. there should be ACP-KS (translocation) between ACP-ER and ACP-AT. The upleft ACP-KS is at condensation stage, but the crosslinking structures should represent translocation stage as I understand.

Response: Revised as suggested. This is now addressed in the caption of **Fig. 1**.

Item 53, Reviewer 2: Fig2b. the linkers’ length shown here do not match the main text such as L147, L162, etc. and the ACP with 300 amino acids in length looks too long.

Response: Revised as suggested. Linker length problems have been addressed in **Fig. 2b** as well as in the main text (line 161).

Item 54, Reviewer 2: ExFig4. Why was Sfp added in the first step, despite its function to be transferring CoA 1 into apo ACP? What is primed 1 exactly? Can crosslinker 1 directly react with ACP, so that R to be OH?

Response: Crystal structures (PDBID: 1QR0 and 4MRT) have shown that CoA binds to Sfp with or without the presence of acyl carrier protein (ACP). By including Sfp in the first step, it allows the CoA 1 (primed 1) to bind to Sfp and to be ready for transferring the CoA 1 to ACP. Crosslinker 1 cannot directly react with ACP. Crosslinker 1 needs to be converted into Coa 1 by CoaA, CoaD, and CoaE followed by the transferring onto the serine of ACP by Sfp. The reason why we label it as “R” in Extended Data Fig. 4 is that MAS complex B does not have enough density to support if crosslinker 1 that inhibits the KS is in the Coa 1 form or the unmodified 1 form (Extended Data Fig. 19).

Item 55, Reviewer 2: ExFig5. Could the authors explain how to interpret multiple crosslinking bands with different sizes?

Response: Revised as suggested. A detailed explanation has been added to the caption of Extended Data Fig. 5.

Item 56, Reviewer 2: ExFig7, 23 and others. What are b1-4?

Response: A detailed explanation has been added to the caption of the Extended Data Figures 7 and 45.

Item 57, Reviewer 2: ExFig22. The use of cerulenin and how to interpret these data are not discussed.

Response: Since the use of cerulenin in Extended Data Fig 22 did not yield conclusive results, we have deleted the bar chart.

Item 58, Reviewer 2: ExFig24 and others. What is the difference between panels A-C?

Response: Extended Data Fig. 23-27 include triplicates of the crosslinking experiments for the bar charts of Extended Data Fig. 22. This has been added to the figure captions.

Item 59, Reviewer 2: ExFig28. The crosslinking between KS and ACP substrate is not visible. The catalytic Cys in KS should be labeled.

Response: Revised as suggested. Extended Data Fig. 28 has been updated with the crosslinking between KS and ACP and labeling for KS reactive residue Cys.

Item 60, Reviewer 3: The lack of discussion of oligomeric state and potentially hydrodynamic radius in solution in general hampers the analysis in several points (see also minor points). For example, when the authors suggest potential mechanisms and conformations that should be supported by experiments beyond a static look at a structure. Indeed, the mutants that the authors characterize may or may not support this mechanism- “the correlation between crosslinking yield and enzymatic activity has been validated”. This is presented by the authors as a general law but is it valid for these mutants that are underpinning the analysis of this specific structure and suggested mechanism? i.e. are these mutations impacting enzymatic activity.

Response: These are excellent points. The enzymes in MAS as well as other Type I FAS such as FASN in humans orchestrate multiple cycles of chain elongation, dehydration, reduction and the impact of a mutant at any position including at the domain interfaces and enzymatic pockets could be very hard to interpret. That said, we have found that evaluating crosslinking efficiency provides

a very accurate tool to interrogate the ability of the ACP to engage with discrete partner protein domains. Unlike a readout of lipid production from the entire synthase, which is often highly inaccurate and difficult to interpret due to variations in product chain length (MAS elongates acyl chains for 1-5 rounds) and oxidation state upon mutation, reports on the ability of the ACP domain to achieve an active state of substrate processing would be difficult to establish by any other means. Moreover, understanding a change in chain length upon mutation would be very hard to explain mechanistically and would in fact provide little insight into the molecular mechanics of protein-protein interactions. This is a very important distinction and one that we have been careful to state clearly in the manuscript. In response, we have added discussion towards these points and find that they provide an improved interpretation of the data and strengthened our discussion on the model.

Item 61, Reviewer 3: In the discussion, the authors should mention and discuss their structure also vis-à-vis the recently published structure of PKS1 and PKS2 from *E. chlorotica* by Schmidt and colleagues (<https://www.biorxiv.org/content/10.1101/2024.07.22.604177v1.full.pdf>) which presents a rocking mechanism for the catalytic cycle.

Response: Revised as suggested. Discussion about the EcPKS1 and EcPKS2 structures has been added to line 411-417.

Item 62, Reviewer 3: A lot of the paper rests on previously established syntheses and site specificity properties. The methods section should include some details about how these were performed and established. E.g. “loading with a one-pot mechanism as previously established.

Response: Revised as suggested. “*MAS crosslinking*” in the method section is now changed to “*MAS crosslinker loading and crosslinking*” with brief description of the one-pot reaction mechanism. More details of the crosslinkers and their specificity are added to the caption of Extended Data Fig. 3.

Item 63, Reviewer 3: The conditions under which the reactions in extended data fig.4 are carried out are not detailed- concentrations, temperature, pH are all missing. I cannot find a methods section corresponding to this priming reaction. Some NMR or mass spec data showing that the priming does indeed generate the expected product would also be beneficial. The meaning of *in situ* in the phrase “we first converted the probes into the primed CoA analogs *in situ*” is also not clear.

Response: Revised as suggested. “The detailed crosslinker loading and crosslinking procedures are described in the general experiment methods section.” has been added to the caption of Extended Data Fig. 4 to guide the readers to the experimental methods for details. The stepwise transformation from crosslinkers to crosslinking CoA analogs has been published previously. [PMID: 17220983] We have inserted the citation in line 69. Since “*in situ*” caused confusion, we have removed it.

Item 64, Reviewer 3: Size exclusion chromatography traces of the sample making up the first structure in the paper and in general size exclusion chromatography traces of final cryo-EM preparations should be shown, indicating which fractions were collected.

Response: Revised as suggested. Size exclusion chromatography traces of the cryo-EM sample are now added in Extended Data Fig. 47 with indication for the fraction being analyzed by cryo-EM.

Item 65, Reviewer 3: Page. 5 “No crosslinking was observed between the cryo-ACP and KS or KS-LD-AT (Extended Data Figs. 5a-b). This is likely due to inactivation of the KS, as it has been

reported that the excised MAS-like PKS KS-LD-AT tridomain construct exists as a monomer-dimer equilibrium in solution.” The authors purified these constructs with size exclusion chromatography and should be able to prove or disprove this statement. The authors should include size exclusion traces with Mw estimates.

Response: Revised as suggested. Size exclusion chromatography traces of truncated MAS KS-LD-AT and MAS KS are now added in Extended Data Fig. 47.

Item 66, Reviewer 3: “The full MAS structure adopts a homodimeric architecture (Fig. 2a), comprising two catalytic chambers (chambers 1 and 2, Fig. 2a)”. Clearly, it is a homodimer in the structure that was solved (complex A), but what was the oligomeric state distribution in solution? SEC or mass photometry data should be shown.

Response: Revised as suggested. Size exclusion chromatography traces and interpretation of the oligomeric states for full MAS as well as crosslinked MAS are now added to Extended Data Fig. 47.

Item 67, Reviewer 3: The crosslinks observed in cryo-EM are likely a small subset of the ones actually formed in solution, i.e. they only comprise those that can be refined. Have the authors considered performing mass spectrometry to characterize the crosslinking, or at least the modification of the various proteins? I would challenge the statement of site-specificity presented in this case, as the authors do not present sufficient evidence that in the case of MAS these crosslinkers are site-specific. I apologize if the evidence is presented in previous work.

Response: Our lab has demonstrated dual site-selectivity of ACP crosslinking in both Type I and Type II FAS and PKS systems. This has been most extensively shown with *E. coli* FAS, with AcpP crosslinked with multiple enzymes (FabB, FabF, FabA, FabZ), as well as in other systems (human mitochondrial FAS and *M. tuberculosis* FASII). Each has been validated to be site-selective via one or multiple forms of MS, NMR, of x-ray crystallography evidence. We are therefore quite confident in these MAS findings. In Extended Data Fig. 6, crosslinkers **2** and **3**, which are analogs to crosslinker **1** with different chain lengths (Extended Data Fig. 3), were loaded onto *E. coli* AcpP and tested against different partner protein including FabF (KS), FabFCys163Ala (KS mutant with reactive residue inactivation), FabG (KR), FabA (DH), FabI (ER), and FabD (AT). AcpP loaded with crosslinkers **2** or **3** was able to crosslink with only FabF and FabA. Similar results were observed in Extended Data Fig. 45. Mutants MAS_{Cys177}Ala (MAS KS reactive residue inactivation) and MAS_{His929}Ala (MAS DH reactive residue inactivation) were tested against crosslinker **1**. Both results demonstrated that crosslinker **1** is site-specific to the KS and DH reactive residues cysteine and histidine respectively.

Item 68, Reviewer 3: The map provided for complex A does not look like a 3.87 angstrom resolution map: the details expected at this resolution are not present in the map. Is the map provided unsharpened? The authors should provide the map-to-model FSC and maybe also a locally filtered map for the higher resolution details in the paper.

Response: The map resolutions were calculated by gold-standard FSC between the two independent half-maps at a FSC cutoff of 0.143. While the edges of the map are less sharp, the interior of these domains, especially the KS domain, are much higher in resolution.

Item 69, Reviewer 3: Looking at the models of complex A and B, secondary structures are not assigned and running dssp shows some strange deformations. This may be due to the authors doing

real space refinement without applying secondary structure restraints. Re-refining in REFMAC or Phenix may improve results.

Response: Thank you for pointing this out. The models have now been re-refined with secondary structure restraints in place.

Item 70, Reviewer 3: Some extended data figures are labelled as “extended data figure 3” and others as “extended data figure S5”

Response: Thanks for catching this. These have been fixed.

Item 71, Reviewer 3: Cloning section should include UNIPROT IDs of the proteins involved.

Response: Revised as suggested. The Cloning Section now includes UniProt IDs.

Item 72, Reviewer 3: Uncropped gel images should be source data, not supplementary figures.

Response: Thank you for this point. We will communicate with the editor to rearrange the uncropped gel images in coordination with publication policies.

Item 73, Reviewer 3: “Since Size exclusion chromatography afforded crosslinked MAS products at >45% crosslinking yield by SDS-PAGE (right, Extended Data Fig. 7).” Meaning not clear.

Response: Revised as suggested. Crosslinking bands in Extended Data Fig. 7 are explained now in the caption.

Item 74, Reviewer 3: I am not sure that the section justifying the choice of C16- α -bromoamide 1 based on *in trans* crosslinking of *E. coli* ACP makes sense: why would this system be an appropriate model for *M. tuberculosis* MAS? I understand the point of conformational flexibility, but isn't *E. coli* ACP chosen simply because of its *in trans* properties?

Response: The *in trans* crosslinking experiments involving MAS and *E. coli* type II FAS systems were designed to emphasize that the PPIs are unique between type I and type II systems. The *in trans* crosslinking study of excised MAS ACP and excised MAS KS or excised tridomain aimed to show that KS domain requires the dimeric form to be functional. The successful crosslinking between excised MAS ACP and full MAS further proved the importance of the dimeric form of KS. Due to the confusion, more detailed explanation has been added to lines 78-87. In terms of the choice of C16- α -bromoamide 1, it was chosen as it mimics the natural substrate (fully reduced long acyl chain) as stated in lines 110-112. Also, the screening of crosslinkers 2, 3, 5, and 6 in *E. coli* (Extended Data Fig. 6) details their ability to crosslink with dehydratase (FabA) which was not discovered before.

Item 75, Reviewer 3: The whole paragraph starting with “Evaluating the resolved crosslinked structures across..” feels more like a discussion section and should perhaps be shortened. At this point, the fact that 3d classification in cryo-EM can provide multiple conformations is not really something that needs mentioning as a novel finding.

Response: Revised as suggested. The paragraph (lines 133-138) is now shortened and changed to “Evaluating these four crosslinked complexes offered several important structural insights. First, dual site-selective crosslinking enabled high-resolution visualization of multidomain synthases detailing MAS ACP interactions with crosslinked domains, adjacent domains, and linker peptides. Second, ACP crosslinking can rigidify architecture of MAS with higher local resolution at the crosslinked sites. Finally, dual site-selective crosslinking greatly enhanced the homogeneity of 4 trapped states within 6 catalytic states in each catalytic chamber.”

Item 76, Reviewer 3: ΨKR abbreviation is introduced before its explanation. In general, reducing the use of domain acronyms would greatly improve readability. Please label domains in figure 1, instead of introducing domain naming/color scheme in figure only to then have the reader look backwards to understand the cycle in Fig.1.

Response: Revised as suggested. We have labeled the domains in Fig. 1. A brief explanation for ΨKR (first instance in main text) has been added to lines 122-123. Domain labels have been added in Fig. 1a.

Item 77, Reviewer 3: “The LD has been previously reported to be involved in a turnstile mechanism, where a closed conformation forms extensive interactions with AT to block access to the KS.” Citation missing

Response: Revised as suggested. This citation has been added.

Nature Communications
NCOMMS-24-72772-T – R2
Response to Reviewer Comments

Item 1, Reviewer 1: One major point remains. The EM map density is ambiguous for the dimer neck that connects the elongation region (KS-LD-AT) to the modifying region (DH-ΨKR-ER-KR-ACP). Thus, how these regions are connected within a protomer cannot be assigned in the structures. Moreover, while the crosslinking data demonstrate that crosslinks can form between protomers (ACP to KS), the data are also consistent with intra-protomer crosslinking. A major band labeled “uncrosslinked” or “b1” is prominent in all denaturing gels of crosslinked protein and likely includes proteins with intra-protomer crosslinks (Fig. 2C & Figs. S7, S23-S27, S45). The mixture of species existed in the sample used for EM.

Response: Apologies for this ambiguity. The problems related to SDS-PAGE gel bands “b1” have been addressed, which illustrate the case intra-protomer crosslinking.

Item 2, Reviewer 1: I suggest a simple relabeling of domains in the text and the figures. The protomers and domains of the MAS dimer are labeled as unprimed or primed (KS, KS', ACP, ACP', etc.). However, it is unknown whether the domains labeled KS-LD-AT are in the same protomer as those labeled DH-ΨKR-ER-KR-ACP or in the protomer labeled DH'-ΨKR'-ER'-KR'-ACP'. If the labels KS-LD-AT and KS'-LD'-AT' are retained for the elongation region of the MAS protomers, then the domains of the modifying region could be labeled DH^A-ΨKR^A-ER^A-KR^A-ACP^A and DH^B-ΨKR^B-ER^B-KR^B-ACP^B. Points in the manuscript file where the inter-protomer interpretation should be changed are highlighted in the file I return with this review. This additionally applies to several of the supplemental figures and captions (10-16, 18-21, 28-29, 40-41, 43, 48, 51-52).

Response: In response to this comment, we changed how the regions are labeled. All prime (') and superscripted letters (^A or ^B) were removed from residue numbers and when needed the domains were listed to highlight the difference between residues and their corresponding domains. The manuscript now has consistent use of:

DH^A instead of DH

DH^B instead of DH'

ΨKR^A instead of ΨKR

ΨKR^B instead of ΨKR'

KR^A instead of KR

KR^B instead of KR'

ER^A instead of ER

ER^B instead of ER'

ACP^A instead of ACP

ACP^B instead of ACP'

KS was left as KS

KS' was left as KS'

LD was left as LD

LD' was left as LD'

AT was left as AT

AT' was left as AT'

Item 3, Reviewer 2: L217-222 The comparison of MAS with modular type I PKSs are provided here, and it would be helpful to explicitly state that the DEBS-M1 and Lsd14 structures are at condensation state (KS-ACP within the same module) . In contrast, the DEBS-M3 and ACP2 structure is at the substrate reloading state (ACP_n and. KS_{n+1}), which is the identical state to the MAS observed structure.

Response: Addressed as suggested. The sentences are now “In comparison, ACPs in modular type I PKSs DEBS-M1 (PDB ID: 7M7F, Extended Data Fig. 20b) and Lsd14 (PDB ID: 7S6C, Extended Data Fig. 20c) adopt slightly different binding modes **at the condensation state**, with more interactions between helix 1 and 2 of ACP and KS, but no interactions with the AT (Extended Data Fig. 20a).^{13,14} However, the observed ACP-AT interactions are in agreement with the recent complex between DEBS-M3 and ACP2 structure **at the transacylation state**.³⁰”

Item 4, Reviewer 2: L389 "allowing the ACP to interact with KS while crosslinking with KS"
Does this describe the two ACPs? The ACP should not interact with KS from the same protomer.

Response: To address this concern, the sentence has been changed to “The cross-protomer processivity plays an important role guiding synthetic efficiency in type I PKSs, and likely FASs by allowing the ACP to interact with KS' while ACP' interacting with KS or *vice versa*. in lines 389-390.

Item 5, Reviewer 2: L436 While I appreciate these discussions about substrate recognition, the TNGQ or VMYH motifs are only useful for modular cis-AT PKSs; for example, in trans-AT PKSs, they do not have these conserved motifs (<https://doi.org/10.1016/j.str.2013.12.016>). Would it be possible to discuss it based on structure? For instance, Fig. 3 shows substrate binding at KS, but it is unclear how the substrate binding pocket (or tunnel) recognizes (or interact with) the alpha,beta-position of the substrate.

Response: We appreciate the differences between cis- and trans-AT PKS systems. The vectorial biosynthesis and enzymology of trans-AT PKSs can differ significantly from those of an iterative cis-AT PKSs. Given that MAS is a cis-AT PKS, it may not be appropriate to directly extrapolate our findings to trans-AT PKSs. For instance, substrates bearing substitutions at the α - or β -position may interact differently with the KS or DH active sites compared to crosslinker 1, which lacks such substitutions. Our current structural model can only offer hypotheses rather than definitive insights on these issues. Therefore, we discuss prior studies (see line 440 in the manuscript) that have motivated us to explore a variety of crosslinkers to capture different catalytic states within the KS domain. This is work that is currently ongoing and falls outside of the scope of this manuscript.

Item 6, Reviewer 3: The authors have addressed my concerns, and I recommend the paper for publication. However, the supplementary figures of the size exclusion chromatography runs have weird artefacts from copy pasting unloaded images, or something like that. They are not displayed properly.

Response: Thank you for pointing this out. Now Extended Data Fig. 47 has been replaced by original chromatograms without cropping or zooming in.